# A Theoretical Framework for Zeroth-Order Budget Convex Optimization

**François Bachoc**                                                                                    *francois.bachoc@math.univ-toulouse.fr*
*Institut de Mathématiques de Toulouse*
*Université Paul Sabatier*
*Institut universitaire de France (IUF)*

**Tommaso Cesari**                                                                                    *tcesari@uottawa.ca*
*School of Electrical Engineering and Computer Science*
*University of Ottawa*

**Roberto Colomboni**                                                                                 *roberto.colomboni@unimi.it*
*Università degli Studi di Milano*
*Politecnico di Milano*

**Andrea Paudice**                                                                                    *apaudice@cs.au.dk*
*Department of Computer Science*
*Aarhus University (AU)*

**Reviewed on OpenReview:** *https://openreview.net/forum?id=bo8vM9j3U0*

## Abstract

This paper studies a natural generalization of the problem of minimizing a convex function $f$ by querying its values sequentially. At each time-step $t$, the optimizer selects a query point $X_t$ and invests a budget $b_t$ (chosen by the environment) to obtain a fuzzy evaluation of $f$ at $X_t$ whose accuracy depends on the amount of budget invested in $X_t$ across times. This setting is motivated by the minimization of objectives whose values can only be determined approximately through lengthy or expensive computations, where it is paramount to recycle past information. In the univariate case, we design ReSearch, an anytime parameter-free algorithm for which we prove near-optimal optimization-error guarantees. Then, we present two applications of our univariate analysis. First, we show how to use ReSearch for stochastic convex optimization, obtaining theoretical and empirical improvements on state-of-the-art benchmarks. Second, we handle the $d$-dimensional budget problem by combining ReSearch with a coordinate descent method, presenting theoretical guarantees and experiments.

## 1 Introduction

Consider the following fundamental question: given a convex real-valued function $f$, how can we efficiently and sequentially select oracle queries of it in order to recommend a point $x$ such that $f(x)$ is as close as possible to the infimum of $f$? This problem is known as zeroth order convex optimization and has been studied for more than half a century (Rosenbrock, 1960). The field has also recently attracted the interest of the machine learning and statistical community because computing the gradient of a function that depends on a large dataset (e.g., the empirical risk) can be very expensive if not unfeasible (see for example Bubeck et al. 2021 and references therein). Another significant application arises in simulation-based optimization, where the goal is to optimally tune the parameters of a system by only observing its output (Conn et al., 2009; Spall, 2005).

There are several different ways to model this problem. In the deterministic setting, the oracle answers each query $x$ with the exact value $f(x)$. The classic stochastic setting alleviates the restrictiveness of the deterministic oracle by assuming that each query $x$ returns a noisy independent estimation of $f(x)$. This oracle model is still not flexible enough to cover applications where perturbations are not independent or where the optimizer can compute the value $f(x)$ at a query point $x$ with incremental precision. The former is crucial to include scenarios where errors have long-range dependence (Lahiri, 2003; Beran, 2017). The latter has several practical applications, e.g., when the values of $f$ are the results of long sums (as in time-series forecasting via weighted empirical risk minimization; Kuznetsov & Mohri 2015; 2016) or, crucially, when they can only be computed approximately through lengthy simulations (as it happens ubiquitously in the field of computer experiments; Santner et al. 2003; Sacks et al. 1989).

**Contributions.**   We make the following contributions:

- We design a novel zeroth-order *budget* optimization setting where the oracle answers each query $x$ with an interval that is guaranteed to contain $f(x)$ and whose length decreases with the amount of budget invested on $x$ so far. In addition to generalizing the deterministic and stochastic settings, our model also captures the aforementioned problems not covered by them. (Section 2.)
- We design ReSearch, an *anytime*, *practicable*, and *parameter free* algorithm for univariate zeroth-order budget convex optimization that works under a minimal convexity assumption on $f$. (Section 3.1.)
- We prove a sharp *anytime* upper bound on the optimization error of ReSearch. Furthermore, our analysis reveals that the optimal dependence on the Lipschitz constant of $f$ is extremely mild, asymptotically negligible, and can be entirely lifted by transitioning to a continuous budget optimization setting. (Section 3.2.)
- We prove a matching (up to constants) lower bound, certifying the optimality of ReSearch. (Section 3.3.)
- We apply ReSearch and its analysis to univariate stochastic convex optimization, improving the state-of-the-art guarantees for this problem. (Section 4.1.)
- We illustrate how to handle the *d*-dimensional budget setting using ReSearch as a subroutine of a coordinate-descent algorithm and provide corresponding theoretical guarantees. (Section 4.2.)
- Finally, we present illustrative experiments supporting our theory in the univariate stochastic and uni/multivariate budget settings. (Section 5.)

**Related Work.**   Zeroth-order convex optimization is a massive field with vast literature. We limit our discussion to references more closely aligned with the scope of this paper.

The deterministic case is the simplest setting in zeroth-order optimization, where the oracle answers each query $x$ with the exact value of the objective $f(x)$ (see Nesterov et al. 2018 and references therein). Although not the core of our work, we highlight that in this setting, our one-dimensional algorithm ReSearch achieves the well-known optimal geometric decay on the optimization error while not requiring the objective to be globally Lipschitz.

To the best of our knowledge, our flexible budget setting with errors decaying as functions of the budget is not addressed theoretically in the convex optimization literature. The more specific stochastic setting, where the oracle answers queries with random independent estimates of the objective, is studied in particular by Agarwal et al. (2013); Jamieson et al. (2012); Shamir (2013); Belloni et al. (2015). Our budget setting recovers it as a special case when errors decay as $O(1/\sqrt{\text{budget}})$. In the one-dimensional case, an optimization error of $\Omega(1/\sqrt{T})$ is unavoidable in the stochastic setting, even knowing the time-horizon $T$ in advance and under the additional assumptions of smoothness and strong-convexity (see Shamir 2013, Theorem 3). This rate is also achieved by Belloni et al. (2015) and Lattimore (2020), with high-probability and in-expectation respectively, up to extra log terms; however, these algorithms are quite involved. Agarwal et al. (2013) and Jamieson et al. (2012) propose simpler and more practical trisection-based algorithms with similar optimization error guarantees. While these algorithms share some features with ReSearch (e.g., they monitor confidence-interval separation to discard domain portions), our analysis departs substantially from those of Agarwal et al. (2013) and Jamieson et al. (2012), leading to additional theoretical benefits (in particular, a negligible dependence on the Lipschitz constant and an improved logarithmic dependence on the time horizon $T$). Finally, in

contrast to the works above, we highlight that our reduction from the univariate budget to the univariate stochastic setting hold without any additional assumption (such as strong convexity, smoothness, or global Lipschitzness).

## 2 Setting

Given a bounded convex set $I \subset \mathbb{R}^d$, our goal is to minimize an unknown *convex* function $f : I \to \mathbb{R}$ picked by a possibly adversarial and adaptive environment by only requesting fuzzy evaluations of $f$. At every interaction $t$, the optimizer selects a query point $X_t$ and the environment selects and reveals a budget $b_t$. This budget is then used to reduce the fuzziness on the value of $f(X_t)$, modeled by an interval $J_t \ni f(X_t)$. In other words, the reader might think of the budget as a perishable (must be spent in full at every interaction) and non-divisible (all must be spent in a single query point) amount of resources made available by the environment to reduce the fuzziness of the value of the unknown objective at the current query point.

The interactions between the optimizer and the environment are described in Optimization Protocol 1.

---
**Optimization Protocol 1**

---
**input:** A non-empty bounded convex set $I \subset \mathbb{R}^d$ (the domain of the unknown objective $f$)

1: **for** $t = 1, 2, \ldots$ **do**
2:   The optimizer selects a query point $X_t \in I$ where to invest the next budget
3:   The environment picks and reveals budget $b_t > 0$ and an interval $J_t \subset \mathbb{R}$ such that $f(X_t) \in J_t$
4:   The optimizer recommends a point $R_t \in I$

---

We stress that the environment is adaptive. Indeed, the intervals $J_t$ that are given as answers to the queries $X_t$ can be chosen by the environment as an arbitrary function of the past history, as long as they represent fuzzy evaluations of the convex objective $f$ (i.e., as long as $f(X_t) \in J_t$ for all $t$).

Note that optimization would be impossible without further restrictions on the behavior of the environment, since an adversarial convex environment could return $J_t = \mathbb{R}$ for all $t \in \mathbb{N}$, making it impossible to gather any meaningful information. We limit the power of the environment by relating the amount of budget invested in a query point $X_t$ with the length of the corresponding fuzzy representation $J_t$ of $f(X_t)$, as quantified by the following assumption.

**Assumption 2.1.** There exist $c \geq 0$ and $\alpha > 0$ such that, for any $t \in \mathbb{N}$, if the optimizer invested the budgets $b_1, \ldots, b_t$ in the query points $X_1, \ldots, X_t$, then

$$|J_t| \leq c/\mathfrak{B}_t^\alpha ,$$

where $|J_t|$ denotes the length of $J_t$ and $\mathfrak{B}_t := \sum_{s=1}^t b_s \mathbb{I}\{X_s = X_t\}$ is the total budget invested in $X_t$ up to time $t$.

The performance of a recommendation $R_T$ is evaluated with the *optimization error* $f(R_T) - \inf_{x \in I} f(x)$.

## 3 An optimal algorithm for the univariate case

In this section we study the univariate budget convex optimization problem, i.e., the case when the underlying convex set $I \subset \mathbb{R}$ is a bounded interval. To solve this problem we propose ReSearch, an algorithm exploiting the 1-dimensional nature of the problem by following a query strategy that allows the learner to recycle most of the past queries (Algorithm 2).

### 3.1 ReSearch

Before presenting its pseudo-code, we introduce some notation. For any positive integer $n \in \mathbb{N}$ we denote by $[n]$ the set $\{1, \ldots, n\}$ of the first $n$ integers. Let $\mathcal{P} := \{\blacksquare\square\square\square, \square\square\square\blacksquare, \blacksquare\blacksquare\square\square, \square\square\blacksquare\blacksquare, \blacksquare\square\square\blacksquare\}$. The blackened parts of the elements of $\mathcal{P}$ represent which portions of the active interval maintained by ReSearch the algorithm

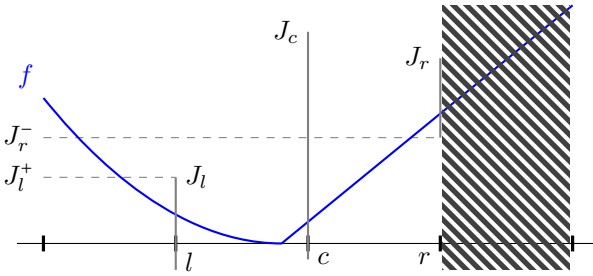

Figure 1: A representation of the delete function. Since $J_l^+ \leq J_r^-$, the points right of $r$ are deleted, i.e., $\mathrm{delete}(J_l, J_c, J_r) = \square\square\square\blacksquare$.

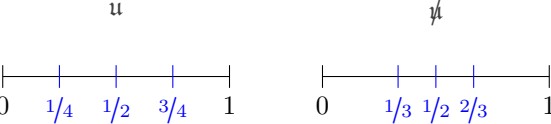

Figure 2: The uniform ($\mathfrak{u}$) and non-uniform ($\mathfrak{n}$) partition functions applied to the interval $I = [0, 1]$.

will delete. Additionally, we will consider the element $\square\square\square\square$ representing the case where no parts of the active interval will be deleted. Let $\mathcal{J}$ be the set of all intervals, and $\mathcal{I} \subset \mathcal{J}$ that of all *bounded* intervals. Furthermore, for any interval $J \in \mathcal{J}$, let $J^- := \inf(J)$ and $J^+ := \sup(J)$. ReSearch relies on four auxiliary functions: the delete function, the uniform partition function $\mathfrak{u}$, the non-uniform partition function $\mathfrak{n}$, and the update function. The delete function (see Figure 1)

$$\mathrm{delete} \colon \mathcal{J}^3 \to \mathcal{P} \cup \{\square\square\square\square\}$$

is defined, for all $(J_l, J_c, J_r) \in \mathcal{J}^3$, by

$$\begin{cases} \blacksquare\blacksquare\square\square & \text{if } J_c^- \geq J_r^+, \text{ else} \\ \square\square\blacksquare\blacksquare & \text{if } J_c^- \geq J_l^+, \text{ else} \\ \blacksquare\square\square\blacksquare & \text{if } J_l^- \geq \min(J_c^+, J_r^+) \ \& \ J_r^- \geq \min(J_l^+, J_c^+), \text{ else} \\ \blacksquare\square\square\square & \text{if } J_l^- \geq \min(J_c^+, J_r^+), \text{ else} \\ \square\square\square\blacksquare & \text{if } J_r^- \geq \min(J_l^+, J_c^+), \text{ else} \\ \square\square\square\square & . \end{cases}$$

In words, the intervals $J_l, J_c, J_r$ will represent the fuzzy evaluations of three points $l < c < r$ in the domain of the unknown objective (left, center, and right). Since we are assuming that the objective is convex, note that whenever an upper bound on the value of the objective at a point $x$ is lower than the lower bound at another point $y$ that is left (resp., right) of $x$, then, all points that are left (resp., right) of $y$ ($y$ included) are no better than $x$. Therefore, the function delete returns which part of an interval containing three distinct points $l < c < r$ should be deleted given the fuzzy evaluations $J_l, J_c, J_r$. (E.g., $\blacksquare\blacksquare\square\square$ represents the deletion of all points of the active interval left of $c$, $\square\square\square\blacksquare$ represents the deletion of all points of the active interval right of $r$, $\square\square\square\square$ is returned when the fuzzy evaluations are not sufficient to delete anything, etc.)

The uniform and non-uniform partition functions (see Figure 2) are defined, respectively, by

$$\mathfrak{u} \colon \mathcal{I} \to \mathbb{R}^3, \quad I \mapsto \left( \tfrac{3}{4} I^- + \tfrac{1}{4} I^+, \tfrac{1}{2} I^- + \tfrac{1}{2} I^+, \tfrac{1}{4} I^- + \tfrac{3}{4} I^+ \right),$$

$$\mathfrak{n} \colon \mathcal{I} \to \mathbb{R}^3, \quad I \mapsto \left( \tfrac{2}{3} I^- + \tfrac{1}{3} I^+, \tfrac{1}{2} I^- + \tfrac{1}{2} I^+, \tfrac{1}{3} I^- + \tfrac{2}{3} I^+ \right).$$

In words, the uniform (resp., non-uniform) partition function $\mathfrak{u}$ (resp., $\mathfrak{n}$) returns the three points that are at $1/4$, $1/2$, and $3/4$ (resp., $1/3$, $1/2$, and $2/3$) of the input interval $I$.

The update function

$$\text{update}\colon \mathcal{I} \times \{\mathrm{u}, \cancel{\mathrm{u}}\} \times \mathcal{P} \to \mathcal{I} \times \{\mathrm{u}, \cancel{\mathrm{u}}\}$$

is defined, for all $(I, \vartheta, \mathrm{del}) \in \mathcal{I} \times \{\mathrm{u}, \cancel{\mathrm{u}}\} \times \mathcal{P}$, by the table:

| | u | ⊮ |
|---|---|---|
| ■■□□ | $\left(\left[\frac{1}{2}I^- + \frac{1}{2}I^+,\, I^+\right], \mathrm{u}\right)$ | $\left(\left[\frac{1}{2}I^- + \frac{1}{2}I^+,\, I^+\right], \cancel{\mathrm{u}}\right)$ |
| □□■■ | $\left(\left[I^-,\, \frac{1}{2}I^- + \frac{1}{2}I^+\right], \mathrm{u}\right)$ | $\left(\left[I^-,\, \frac{1}{2}I^- + \frac{1}{2}I^+\right], \cancel{\mathrm{u}}\right)$ |
| ■■■□ | $\left(\left[\frac{3I^- + I^+}{4},\, \frac{I^- + 3I^+}{4}\right], \mathrm{u}\right)$ | $\left(\left[\frac{2I^- + I^+}{3},\, \frac{I^- + 2I^+}{3}\right], \mathrm{u}\right)$ |
| ■□□□ | $\left(\left[\frac{3}{4}I^- + \frac{1}{4}I^+,\, I^+\right], \cancel{\mathrm{u}}\right)$ | $\left(\left[\frac{2}{3}I^- + \frac{1}{3}I^+,\, I^+\right], \mathrm{u}\right)$ |
| □□□■ | $\left(\left[I^-,\, \frac{1}{4}I^- + \frac{3}{4}I^+\right], \cancel{\mathrm{u}}\right)$ | $\left(\left[I^-,\, \frac{1}{3}I^- + \frac{2}{3}I^+\right], \mathrm{u}\right)$ |

In words, when applied to an interval $I$, a type of partition $\vartheta$, and a symbol del (representing the subset of $I$ to be deleted), the update function returns as the first component the interval $I$ pruned of the subset of $I$ specified by $\vartheta$ and del, and, as the second component, how the new interval will be partitioned. It can be seen that the types of partitions returned by update are chosen so that our ReSearch algorithm will only query points on a (rescaled) dyadic mesh (e.g., if $I = [0, 1]$, ReSearch will only query points of the form $k/2^h$, for $k, h \in \mathbb{N}$).

For all $t \in \mathbb{N}$, if the sequence of budgets picked by the environment up to time $t$ is $b_1, \ldots, b_t$, the sequence of query points selected by the optimizer is $X_1, \ldots, X_t$, the corresponding feedback is $J_1, \ldots, J_t$ (see Optimization Protocol 1), then, for each $x \in \mathbb{R}$, we define the quantities

$$\mathfrak{B}_{x,t} \coloneqq \sum_{s=1}^{t} b_s \mathbb{I}\{X_s = x\} \qquad \text{and} \qquad J_{x,t} \coloneqq \bigcap_{s \in [t], X_s = x} J_s$$

with the understanding that $J_{x,t} = \mathbb{R}$ whenever $X_s \ne x$ for all $s \in [t]$. Furthermore, define $\mathfrak{B}_{x,0} = 0$ for all $x \in \mathbb{R}$. In words, $\mathfrak{B}_{x,t}$ is the total budget that has been invested in $x$ by the optimizer up to and including time $t$, while $J_{x,t}$ is the best fuzzy evaluation of the unknown objective at $x$ that is available at the end of time $t$.

The pseudocode of ReSearch is provided in Algorithm 2. ReSearch proceeds in epochs $\tau$ where it maintains an active interval $I_\tau$ and three query points $l_\tau, c_\tau, r_\tau \in I_\tau$. During each epoch $\tau$, it repeatedly queries a point in $\{l_\tau, c_\tau, r_\tau\}$ where it invested the least amount of budget until the function delete has gathered enough information to prune the current active interval. When this happens, first it updates the active interval and the type of partition using the update function. Then, it computes the three query points $l_{\tau+1}, c_{\tau+1}, r_{\tau+1}$ of the next epoch $\tau + 1$. Notably, among $l_{\tau+1}, c_{\tau+1}, r_{\tau+1}$ there will be the point among $l_\tau, c_\tau, r_\tau$ that has the smallest value of $f$ (or one of them, if there are more than one). Afterwards, the algorithm recommends a point $x \in \{l_{\tau+1}, c_{\tau+1}, r_{\tau+1}\}$ with the best known upper bound $J_{x,t}^+$ on the value of $f(x)$ available at the present time $t$,[1] and concludes the current epoch. In all rounds in which function delete has not yet gathered enough information to prune the current active interval, the algorithm makes different recommendations depending on whether or not the amount of budget invested in the current epoch is higher than the amount of budget spent in all past epochs combined. See Figure 3 for an illustration of how ReSearch works.

We stress that ReSearch is any-time (it does not need to know the time horizon $T$ *a priori*), any-budget (it does not need to know the total budget $B \coloneqq \sum_{t=1}^{T} b_t$) and does not require the unknown objective to be Lipschitz. Nevertheless, we will show in Theorems 3.1 and 3.2 that its performance is guaranteed to be near-optimal even when compared to algorithms with full knowledge of $T$ and $B$, and run on convex Lipschitz functions with *known* Lipschitz constant.

## 3.2 Upper bound

We now provide theoretical guarantees for ReSearch.

---

[1]Under Assumption 2.1, this corresponds to recommending a point $x \in \{l_\tau, c_\tau, r_\tau\}$ with the best known upper bound $J_{x,t}^+$ on the value of $f(x)$ that will "survive" as a query point of the next epoch. Indeed, for $x \in \{l_{\tau+1}, c_{\tau+1}, r_{\tau+1}\} \smallsetminus \{l_\tau, c_\tau, r_\tau\}$, we have $J_{x,t}^+ = +\infty$, since $x$ has never been evaluated. On the other hand, any $x \in \{l_\tau, c_\tau, r_\tau\}$ has already been evaluated, hence $J_{x,t}^+ < \infty$.

---

**Algorithm 2** ReSearch

---

**input:** A non-empty bounded interval $I \subset \mathbb{R}$ (the domain of the unknown objective)
**initialization:** $I_1 \coloneqq [I^-, I^+]$, $\vartheta_1 \coloneqq \mathfrak{u}$, $(l_1, c_1, r_1) \coloneqq \vartheta_1(I_1)$, $t_0 \coloneqq 0$, $B_0 \coloneqq 0$, $B_{1,0} \coloneqq 0$

1: **for** epochs $\tau = 1, 2, \ldots$ **do**
2:     **for** $t = t_{\tau-1} + 1, t_{\tau-1} + 2, \ldots$ **do**
3:         Query $X_t \in \mathrm{argmin}_{x \in \{l_\tau, c_\tau, r_\tau\}} \mathfrak{B}_{x,t-1}$
4:         Let $\mathrm{del}_t \coloneqq \mathrm{delete}(J_{l_\tau,t}, J_{c_\tau,t}, J_{r_\tau,t})$
5:         Let $B_{\tau,t} \coloneqq B_{\tau,t-1} + b_t$ and $\tau_t \coloneqq \tau$
6:         **if** $\mathrm{del}_t \neq \square\square\square\square$ **then**
7:             Let $t_\tau \coloneqq t$, $B_\tau \coloneqq B_{\tau,t}$, and $B_{\tau+1,t} \coloneqq 0$
8:             Let $(I_{\tau+1}, \vartheta_{\tau+1}) \coloneqq \mathrm{update}(I_\tau, \vartheta_\tau, \mathrm{del}_t)$
9:             Let $(l_{\tau+1}, c_{\tau+1}, r_{\tau+1}) \coloneqq \vartheta_{\tau+1}(I_{\tau+1})$
10:        Recommend $R_t \in \mathrm{argmin}_{x \in \{l_{\tau+1}, c_{\tau+1}, r_{\tau+1}\}} J_{x,t}^+$
11:        **break**
12:       **else if** $B_{\tau,t} \geq \sum_{\tau'=0}^{\tau-1} B_{\tau'}$ **then**
13:        Recommend $R_t \in \mathrm{argmin}_{x \in \{l_\tau, c_\tau, r_\tau\}} J_{x,t}^+$
14:       **else**
15:        Recommend $R_t \coloneqq R_{t_{\tau-1}}$

---

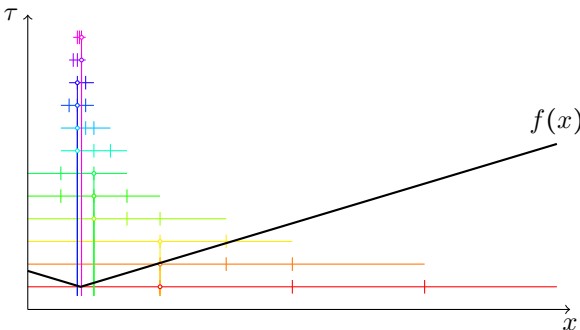

Figure 3: A run of ReSearch. Here, the function is piece-wise linear and its graph is in thick black. The horizontal segments are the active intervals $I_\tau$ of consecutive epochs $\tau$. The short vertical segments are the current query points $l_\tau, c_\tau, r_\tau$ of epoch $\tau$, and the dots (prolonged down vertically) are the recommendations at the end of each epoch, that converge towards $x^\star$. Note that, from one epoch to the next, two out of three points are kept (together with their guarantees), maximizing the recycling of past information.

**Theorem 3.1.** *For any bounded interval $I \subset \mathbb{R}$, if the optimizer is running ReSearch (Algorithm 2) with input $I$ in an environment satisfying Assumption 2.1 for some $c \geq 0$ and $\alpha > 0$, then, there exist $c_1 \leq 12 \cdot \left(48/(2^{1/\alpha} - 1)\right)^\alpha, c_2 \leq 9/8, c_3 \geq (\ln 2)/48$ such that, for any time $T \in \mathbb{N}$, every sequence of budgets $b_1, \ldots, b_T > 0$, and every convex function $f \colon I \to \mathbb{R}$, the optimization error $f(R_T) - \inf_{x \in I} f(x)$ is upper bounded by*

$$c_1 \cdot \frac{c}{(\sum_{t=1}^T b_t)^\alpha} + c_2 \cdot L \, |I| \exp\left(-c_3 \cdot \frac{\sum_{t=1}^T b_t}{\max_{t \in [T]} b_t}\right), \tag{1}$$

*where $L$ is the local Lipschitz constant of $f$ on $[l_{\tau_T}, r_{\tau_T}]$.*

The full proof of this result can be found in Appendix A. Before presenting a sketch of it here, we make a few remarks. First, note that the bound is non-trivial even when the function is *not* globally Lipschitz (as it is the case, e.g., for the function $f(x) = -\sqrt{1 - x^2}$ defined on the interval $I = [-1, 1]$), since it depends on a *local* Lipschitz constant $L$ (which is always finite) that, informally, as the epochs go by, captures better and better how much the function varies around the points that are close to the minimum.[2] Second, note that (up to the constants $c_1, c_2, c_3$) the bound consists of two terms.

The first term $c/(\sum_{t=1}^T b_t)^\alpha$ is a consequence of the fuzziness of the evaluations, that is regulated by Assumption 2.1: when $\sum_{t=1}^T b_t \geq 1$, it decreases when $\alpha$ increases or $c$ decreases. Moreover, when $c = 0$ and $b_t = 1$ for all $t \in [T]$, our problem reduces to deterministic convex optimization. In this case, the first term vanishes completely, leaving behind only the known optimal exponentially-decaying rate $L |I| e^{-\Omega(T)}$ for deterministic convex optimization.

The second term $L |I| e^{-\Omega(\sum_{t=1}^T b_t / \max_{t \in [T]} b_t)}$ is a consequence of the discrete nature of our setting. Notably, if the optimizer could choose to invest infinitesimally small budgets $b_t$ (i.e., if the discrete optimization protocol became a continuous one), the term would vanish completely. Strikingly, when this is the case, the bound becomes completely *independent* of the Lipschitz constant $L$. To the best of our knowledge, this is the first result in convex optimization that shows how the dependence on $L$ could be *entirely* lifted if we transitioned from a discrete to a continuous setting. In other words, our bound gives a parameterization of the dependence on the Lipschitz constant in terms of how close our setting is to a continuous one. The high-level reason for this behavior is that, in a discrete setting, the optimizer might be forced to spend a large amount of budget $b_t$ on a point $X_t$ where a significantly smaller investment would have been sufficient to determine whether or not that point was suboptimal. In this case, if the function is varying significantly, the number of queries could not be sufficient to get close to a minimizer, and this would yield an optimization error that scales with $L$. Finally, we note that, naturally, the Lipschitz constant $L$ and the domain length $|I|$ appear as a product. Indeed, shrinking (resp., dilating) the domain of a function $f \colon I \to \mathbb{R}$ corresponds (inversely-proportionally) to an increase (resp., decrease) of the Lipschitz constant.

*Proof sketch.* We divide the analysis in the 3 cases sketched below, depending on how ReSearch selects $R_T$.

1. $\mathrm{del}_T \neq \square\square\square\square$. In this case, we partition the number of epochs in several classes and focus our attention on the class where we invested the highest fraction of the total budget $\sum_{t=1}^T b_t$. Say that this class contains $n$ epochs. If $n$ is small, we show that in the last epoch of this class there exist two query points that are near-optimal and that the recommendation $R_T$ of ReSearch has guarantees that are close to those of these two near-optimal points. If, on the other hand, $n$ is large, the result follows by the local Lipschitzness of $f$.

2. $\mathrm{del}_T = \square\square\square\square$ and the majority of the budget was invested in the last epoch. In this case, we split again the analysis in two further cases. If the maximum budget $\max_{t \in [T]} b_t$ is small, we show that all three query points of the last epoch are near-optimal, therefore so is the recommendation $R_T$. If, on the other hand, the maximum budget $\max_{t \in [T]} b_t$ is large, we fall back again to the local Lipschitzness of the objective.

3. $\mathrm{del}_T = \square\square\square\square$ and the majority of the budget was invested before the last epoch. Since in this case the recommendation $R_T$ is the same as the recommendation that ended the previous epoch, the result follows by Item 1, using half of the total budget. □

---

[2]For more on the advantages of an adaptive local Lipschitz constant, see Appendix E.

### 3.3 Lower bound

In this section, we show that ReSearch is worst-case optimal: there exist instances where the upper bound of Theorem 3.1 is matched (up to possibly different constants $c_1, c_2, c_3$). The apparent asymmetry between our upper and lower bounds is due to the fact that, in Theorem 3.2:

- We gave the optimizer the freedom to select the time horizon $T$ and total budget $B$ ahead of time.
- We restricted the result to convex Lipschitz functions.

Note that both these changes make our results stronger, since ReSearch is able to match the lower bound despite lacking the freedom to select $T, B$ (in fact, being totally oblivious to a possibly adversarial choice of both) and Theorem 3.1 holds even for non-Lipschitz functions.

**Theorem 3.2.** *For any nondegenerate bounded interval $I \subset \mathbb{R}$, if the environment satisfies Assumption 2.1 for some $c \geq 0$ and $\alpha > 0$, then, there exist $c_1 \geq 1/4, c_2 \geq 1/32e, c_3 \leq 1$ such that, for any time $T \in \mathbb{N}$, every total budget $B > 0$, every Lipschitz constant $L > 0$, and every deterministic algorithm run by the optimizer, there exists a sequence of budgets $b_1, \ldots, b_T$ such that $\sum_{t=1}^{T} b_t = B$ and there exists a $\max\left(\frac{c}{|I|B^\alpha}, L\right)$-Lipschitz convex function $f: I \to \mathbb{R}$, for which the optimization error $f(R_T) - \inf_{x \in I} f(x)$ is lower bounded by*

$$c_1 \cdot \frac{c}{(\sum_{t=1}^{T} b_t)^\alpha} + c_2 \cdot L |I| \exp\left(-c_3 \cdot \frac{\sum_{t=1}^{T} b_t}{\max_{t \in [T]} b_t}\right) . \tag{2}$$

We defer the proof of this result to Appendix B.

## 4 Applications

We present two notable applications of our method. First, we show how to apply ReSearch to the case of univariate stochastic convex optimization, improving on state-of-art bounds; remarkably and in contrast with previous works, the algorithm does not require the Lipschitzness of the objective. Second, we illustrate how to address the multivariate budget case by combining ReSearch with a classic coordinate descent method (see Tseng 2001 and references therein) when the objective is smooth and strongly convex.

### 4.1 Univariate Stochastic Convex Optimization

In this section, we show how to apply ReSearch to the related problem of univariate stochastic convex optimization (SCO). Typically, in this problem, one assumes that querying a point $x$ returns an i.i.d. subgaussian (noisy) evaluation of the unknown objective $f(x)$. Instead, we will introduce a more general setting where the key property is the concentration of the (averages of the) queried evaluations. This way, we can recover the classic SCO but also obtain results for more general non-i.i.d. settings (see below).

Let $I$ be a bounded interval and $f: I \to \mathbb{R}$ an unknown convex function.

**Assumption 4.1.** There exist $\alpha > 0$, $c: (0, 1) \to (0, \infty)$, $m: \bigcup_{t \in \mathbb{N}} \mathbb{R}^t \to \mathbb{R}$, and a family of random variables $(Y_{x,s})_{x \in I, s \in \mathbb{N}}$ safisfying, for all $x \in I$ and $t \in \mathbb{N}$,

$$\mathbb{P}\left[\left|m(Y_{x,1}, \ldots, Y_{x,t}) - f(x)\right| \leq \frac{1}{2} \frac{c(\delta)}{t^\alpha}\right] \geq 1 - \delta .$$

Note that, in classic SCO, where for each $x \in I$, the sequence $(Y_{x,t})_{t \in \mathbb{N}}$ is i.i.d. and $\sigma$-subgaussian, Assumption 4.1 is implied by Hoeffding's inequality for $\alpha := 1/2$, $c(\delta) := \sqrt{8\sigma^2 \ln(2/\delta)}$ (for all $\delta \in (0, 1)$) and $m$ as the empirical average. The case $\alpha < 1/2$ in Assumption 4.1 is relevant to model errors with long-range dependence (Lahiri, 2003; Beran, 2017). To give a simple example, consider that $(Y_{x,t})_{t \in \mathbb{N}}$ are Gaussian with $\text{Cov}(Y_{x,t}, Y_{x,s}) = \sigma^2(1 + |t - s|)^{-\beta}$ for $t, s \in \mathbb{N}$ and for a fixed $\beta \in (0, 1)$. Then, it can be checked that Assumption 4.1 holds with $m$ as the empirical average, with $\alpha := \beta/2$ and with $c(\delta) := \sqrt{16\sigma^2 \ln(2/\delta)}$. This is obtained by bounding the variance and then the tail of the (Gaussian) average.

Let $T \in \mathbb{N}$ be the time horizon. The learner interacts with the environment according to Optimization Protocol 3. As before, the goal is to minimize the optimization error after $T$ time steps $f(R_T) - \inf_{x \in I} f(x)$.

---

**Stochastic Optimization Protocol 3**

---

1: **for** $t = 1, \ldots, T$ **do**
2:     The optimizer selects a query point $X_t \in I$
3:     The environment reveals $Y_{X_t, N_t}$, where
$$N_t \coloneqq \sum_{s \in [t]} \mathbb{I}\{X_s = X_t\}$$
4: The optimizer recommends a point $R_T \in I$

---

By running ReSearch with feedback $J_t$ equal to a suitable confidence interval for $f(X_t)$ (at each time $t$), we extend the state-of-the-art for stochastic convex optimization (Agarwal et al., 2013) beyond the globally-Lipschitz case and, even where the previous guarantees held, we improve them in two ways: we remove a logarithmic factor from the bound and we only pay the Lipschitz constant in the non-dominating term, which decreases exponentially with $T$.

To lighten the notation, let $M_t \coloneqq m(Y_{X_t,1}, \ldots, Y_{X_t, N_t})$ and consider Algorithm 4.

---

**Algorithm 4** ReSearch for SCO

---

**input:** Confidence parameter $\delta \in (0, 1)$

1: **for** $t = 1, \ldots, T$ **do**
2:     ReSearch selects the next query point $X_t$
3:     The optimizer feeds ReSearch with the feedback
$$J_t \coloneqq \left[ M_t - \frac{1}{2} \frac{c(\delta)}{N_t^\alpha}, \ M_t + \frac{1}{2} \frac{c(\delta)}{N_t^\alpha} \right]$$
4: ReSearch recommends a point $R_T$

---

Note also that $M_t$ is (in general) a biased estimate of $f(X_t)$, even in the case when $m$ is the empirical average[3].

**Theorem 4.2.** *If the optimizer runs ReSearch for SCO (Algorithm 4), its optimization error is upper bounded by*

$$c_1 \cdot \frac{c(\delta)}{T^\alpha} + c_2 \cdot L |I| \exp\left( -c_3 \cdot T \right) ,$$

*on the complement of an event having probability $O(T^2 \delta)$, where $c_1, c_2, c_3, L$ are as in Theorem 3.1.*

In particular, in the i.i.d. subgaussian setting, picking $\delta = \Theta(1/T^{5/2})$ yields an expected optimization error of order $\sqrt{(\log T)/T}$, improving the state-of-the-art in Agarwal et al. (2013) by a $\log T$ factor.

We defer the proof of Theorem 4.2 to Appendix C.

## 4.2 Multivariate Budget Convex Optimization

We now illustrate how to use ReSearch to address the multivariate budget setting (see Algorithm 5), where the objective $f$ is defined on a convex bounded subset $I \subset \mathbb{R}^d$. We use an adaptation to the budget setting of a coordinate descent method in the spirit of Jamieson et al. (2012). Coordinate descent is typically analyzed assuming that the objective $f$ is strongly convex and smooth, which we also assume until the end of the section.

Algorithm 5 proceeds by performing sequential line searches. During line search $k$, it uses ReSearch along a segment determined by a base point $x_k$ and a randomly drawn axis $i(k)$, recommending points based on the recommendations of ReSearch. A line search is concluded as soon as the length of the active interval of ReSearch is $\leq \eta$. At the end of each line search, the base point $x_{k+1}$ is updated using the best point found by ReSearch.

---

[3]Because ReSearch recycles past observations multiple times, even after they have been used to decide which part of the current interval to discard. (At a high level, the estimates of the values of the function at the points that we keep have the tendency to "look better".) This recycling of information is a feature, not a flaw of the algorithm, as it allows reaching an approximate minimizer with a smaller number of queries (see Section 5.2).

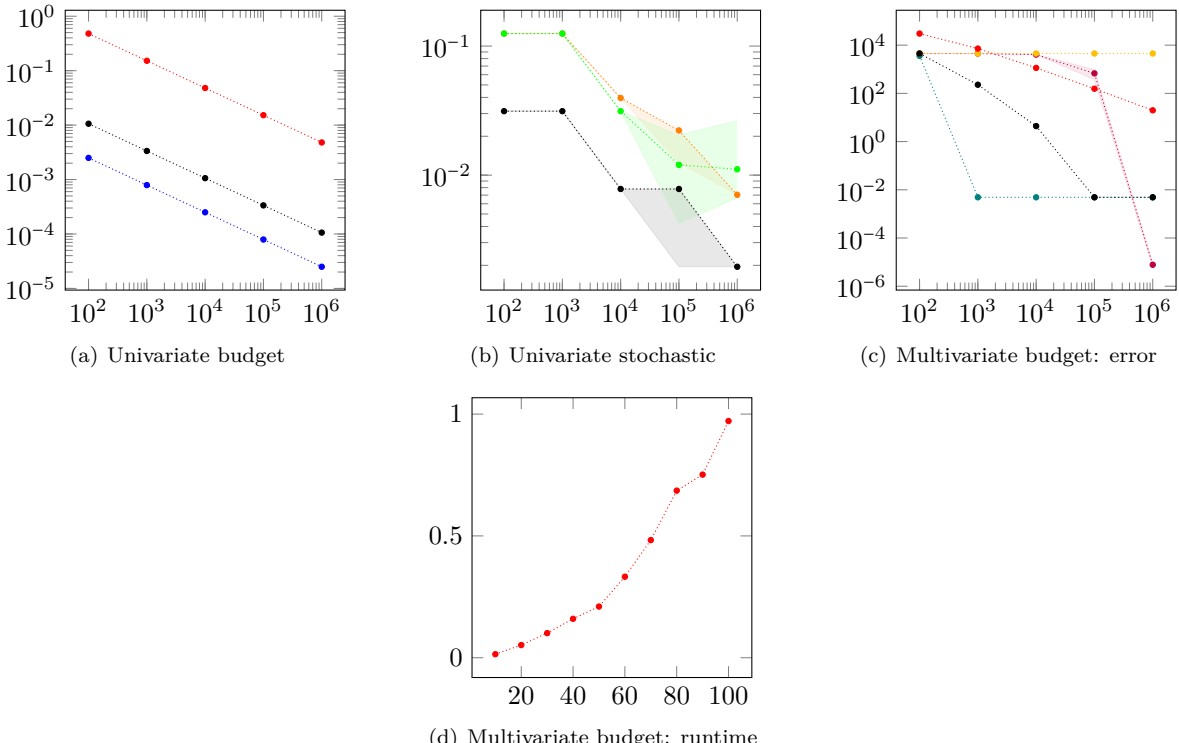

(a) Univariate budget     (b) Univariate stochastic     (c) Multivariate budget: error

(d) Multivariate budget: runtime

Figure 4: (a) The optimization error of ReSearch (black) is compared with the corresponding upper (red) and lower (blue) bounds. (b) The median optimization error of ReSearch for SCO (black) is compared with Jamieson et al. 2012 (green) and Agarwal et al. 2013 (orange); shaded areas are Inter-Quartile Ranges. (c) The average optimization error of Algorithm 5 tuned as in Theorems 4.3 and D.1 (black) is compared with its corresponding upper bound (red) and alternative choices of $\eta$: 1 (blue), 0.1 (purple), 0.01 (orange) —see Section 5.3. Plots are in log scale. (d) The average runtime of Algorithm 5 is plotted against the dimension of the problem —see Section 5.3. All, but (d), plots are in log scale.

---

**Algorithm 5** Budget coordinate descent via ReSearch

**input:** Initial base point $x_0 \in I$, threshold length $\eta > 0$

1: **for** $k = 0, 1, \ldots$ **do**
2:     Pick $i(k)$ uniformly at random from $[d]$
3:     Run ReSearch for time steps $s = 1, \ldots, T_k(\eta)$ on the interval $I_k := \{z \in \mathbb{R} : x_k + z e_{i(k)} \in I\}$, recommending $x_k + R_s e_{i(k)}$ at time $\sum_{j=0}^{k-1} T_j(\eta) + s$, where $T_k(\eta)$ is the first time step where the length of the active interval maintained by ReSearch is $\leq \eta$
4:     Set $x_{k+1} = x_k + R_{T_k(\eta)} e_{i(k)}$

---

For the sake of simplicity, we present our results in the case where the budgets $b_t$ are equal to 1, for all $t$.

**Theorem 4.3** (Informal statement)**.** *Under Assumption 2.1, whenever $f$ is strongly convex and smooth, for any $\eta > 0$, the expected optimization error of Algorithm 5 is upper bounded by*

$$\left(f(x_0) - \inf_{x \in I} f(x)\right)\left(1 - 1/\Theta(d)\right)^{K(T,\eta)} + \Theta(d\eta^2) \,,$$

*where $K(T, \eta)$ is the total number of line searches performed up to time $T$ and the expectation is taken with respect to the random draws of the directions $i(0), i(1), \ldots$.*

*Moreover, an appropriate tuning of $\eta$ yields the bound $\tilde{O}\left(d(d/T)^{\alpha}\right)$ on the expected optimization error.*

For a formal statement, the proof of the previous theorem, and a more general version holding for arbitrary budgets see Appendix D.

We now make the following comments.

First, to put things into perspective, when $\alpha = 1/2$, we note that our rate $d(d/T)^{1/2}$ for the budget setting is in line with state-of-the-art bounds in the related classic field of (i.i.d. subgaussian) stochastic convex optimization (Jamieson et al., 2012), where confidence intervals shrink at the rate in Assumption 2.1, with $\alpha = 1/2$.

Second, in convex optimization, strong convexity usually allows for unconstrained optimization (i.e., $I \coloneqq \mathbb{R}^d$); our method can be extended to this case by adding a pre-processing step before each line search that works by doubling the search space sequentially on the given line until we can guarantee to contain the minimizer. For the sake of conciseness, we leave this standard step out of our presentation.

Third, if one sets a target optimization error of $\varepsilon$, it is a consequence of Theorem D.1 that the runtime of Algorithm 5 is of the order of $(d^{1+\alpha}/\varepsilon)^{1/\alpha}$. This is due to the fact that Line 3 takes constant time for each function query and that after $T$ iterations of Algorithm 5, the optimization error is of the order of $d^{1+\alpha}/T^\alpha$ (ignoring logarithmic factors). Furthermore, the algorithm, at each iteration, only needs to store three points for ReSearch, the $d$-dimensional vector with the current direction, and finally the $d$ pairs of interval limits delimiting the box-constraint. Thus, the overall space complexity is of the order of $d$. We believe that it is unlikely that in this setting one can obtains bounds that are independent from $d$. Indeed, in the related stochastic case, (Shamir, 2013) shows that at least a linear dependence on $d$ is unavoidable.

Finally, we highlight a remarkable benefit of coordinate descent frameworks (in particular, Algorithm 5): they trade off some generality (by requiring strong convexity and smoothness) to gain an easy implementation and obtain efficiency on the two separate fronts of query (by featuring state-of-the-art optimization error bounds) and computational complexity (having low memory requirements and fast execution, irrespectively of the dimension).

## 5 Experiments

We present a preliminary experimental evaluation in support of our theoretical findings, illustrating the following. First, the performance of ReSearch is in line with the theoretical guarantees and, in practice, closer to the lower (Theorem 3.2) than to the upper bound (Theorem 3.1). Second, Algorithm 4 significantly outperforms its natural competitors. Third, the role played by the threshold parameter $\eta$ in Algorithm 5, and its performance compared to the upper bound.

### 5.1 Comparison with upper and lower bounds

This experiment aims at comparing the performance of ReSearch with the corresponding theoretical upper (Theorem 3.1) and lower (Theorem 3.2) bounds. The oracle is based on the setting described in the first part of the proof of Theorem 3.2 with parameters $c = 0.1, \alpha = 1/2, b_t = 1$ for all $t$. We test the performance of ReSearch on a grid of time horizons $T \in \{10^2, 10^3, \ldots, 10^6\}$ against the objective $f$ constructed in the lower bound. Figure 4(a) shows that ReSearch performs well, appearing significantly (note the log scale) closer to the lower than the upper bound.

### 5.2 Stochastic Case

This experiment (Figure 4(b)) aims at showing the effectiveness of Algorithm 4 when compared to the uni-dimensional algorithms proposed in Agarwal et al. (2013) and Jamieson et al. (2012), which are state-of-the-art for this setting. To this end, we optimize the function $f(x) = \frac{1}{2}x^2$ over $I = [0, 1]$; notice that $f$ is 1-Lipschitz over $I$ and its minimum value is 0. The stochastic oracle is implemented according to Stochastic Optimization Protocol 3 with independent Gaussian noise with mean 0 and variance $\sigma^2 = 0.1$. This corresponds to having $c(\delta) = \sqrt{\sigma^2 \ln(1/\delta)}$ and $\alpha = 1/2$ in Assumption 4.1. We set all the algorithms with the parameters indicated by the theory, so that the overall confidence is $1 - 1/T$ in all cases. Consistently

with the previous section, we evaluate the algorithms on a grid of time horizons $T \in \{10^2, 10^3, \dots, 10^6\}$. Since we are comparing high probability bounds, we measure the median optimization error and plot it with the corresponding Inter-Quartile Range over 10 repetitions.

### 5.3 Multivariate case

These experiments illustrate the performance of Assumption 5 in the multivariate budget case. We minimize the function $f(x) = \frac{1}{2}\|x\|_2^2$ over $[-1, 3d]^d$ for $d = 10$, which has the minimum close to a corner, but not exactly there. We implemented the oracle according to Optimization Protocol 1 and Assumption 2.1 with $c = 1, \alpha = 1$ and $b_t = 1$ for all $t$. In addition to using the threshold $\eta_T$ recommended by the tuning in appendix D and the corresponding theoretical upper bound, we also run algorithm 5 using $\eta \in \{1, 10^{-1}, 10^{-2}\}$ for $T \in \{10^2, 10^3, \dots, 10^6\}$. We repeat each run 10 times and report the average optimization error and the standard deviation (note the high concentration: paths have little to no variance). Figure 4(c) shows the results (in logarithmic scale), highlighting that the tuning of the threshold $\eta_T$ dictated by the theory can be a little more conservative in some cases than some *ad hoc* choices of $\eta$, but it is still far better than the upper bound (which is on par with state-of-the-art bounds for analogous settings) and a random guess of $\eta$.

In a second experiment, we considered the same problem setting and measured the runtime of the algorithm against the dimension, varying the dimension $d$ in the interval $[10, 100]$ with a step of 10. We set a target optimization error 0.1 and run the algorithm until it reaches it. In Figure 4(d) we reported the average runtime across 10 repetitions. As predicted by the theory, the runtime follows the predicted $d^2$ behavior and remarkably the execution for $d = 100$ takes less than a second.

## 6 Conclusions

We designed and studied a flexible zeroth-order convex optimization setting, where the accuracy of the queries improves with the invested budget (Assumption 2.1). This framework grants additional modeling power (see Section 1) compared to standard stochastic settings. In dimension one, we designed an any-time, any-budget, parameter-free algorithm, called ReSearch (Algorithm 2), that does not require the *a priori* knowledge of the (local) Lipschitz constant of the objective and works even in an adversarial and adaptive environment. We provided upper and lower bounds on the optimization error that match up to constants. We provided a natural adaptation of ReSearch to the stochastic setting together with a corresponding upper bound featuring the same mild dependence on the Lipschitz constant, and further improved the state-of-the-art for this setting by a logarithmic term. Finally, in the multivariate budget setting, we illustrated the benefits of using ReSearch as a line search procedure in a coordinate descent method: this results in a numerically efficient and practicable budget optimizer that can run even in high-dimensional spaces.

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

# A    Full proof of Theorem 3.1

In this section, we give a detailed proof of Theorem 3.1.

*Proof of Theorem 3.1.* Fix any bounded interval $I \subset \mathbb{R}$, a time $T \in \mathbb{N}$, and a convex function $f \colon I \to \mathbb{R}$. Without loss of generality, we can (and do!) assume that $I$ contains at least two distinct points.[4] Moreover, without loss of generality, we can also (and do!) assume that $f$ attains its minimum in $I$.[5] Then, note that the active interval $I_\tau$ of any epoch $\tau \in [\tau_T]$ (defined in the initialization and updated at Line 8) always contains at least a minimizer, because the first active interval $I_1$ is the entire domain $I$ and, by the unimodality of $f$ and the definition of the delete and update functions, ReSearch deletes a fraction of the active interval (Line 8) only if it is certain that the value of $f$ at one of the remaining points is no bigger than *all* of the deleted points. Thus, there exists (and we fix for the rest of the proof) an $x^\star \in I$ such that $x^\star \in I_{\tau_T} \subset \ldots \subset I_1$ and $f(x^\star) = \min(f)$. Recall that, for all $t \in \mathbb{N}$, $\tau_t$ is the epoch of round $t$ (Line 5 of Algorithm 2). Also, for the sake of convenience, we define $t_0 \coloneqq 0$ and, if the last epoch is not concluded exactly at the end of time $T$, we redefine $t_{\tau_T} \coloneqq T$ and $B_{\tau_T} \coloneqq B_{\tau_T, T}$.

To prove the result, we analyze separately the performance of the recommendation $R_T$ of ReSearch in the three cases of Lines 10, 13 and 15 in Algorithm 2.

Assume at first that $\mathrm{del}_T \neq \square\square\square\square$ (i.e., the condition on Line 6 is true and we recommend $R_T$ as in Line 10). We partition the epochs $\tau \in [\tau_T]$ into four sets, depending on whether or not the epoch is uniform and whether or not $x^\star \leq c_\tau$. More precisely, for any $\zeta \in \{u, \not u\}$ and $\vdash \in \{\leq, >\}$, we let $A_{\zeta, \vdash}$ be the set of all epochs $\tau \in [\tau_T]$ such that $\vartheta_\tau = \zeta$ and $x^\star \vdash c_\tau$. (or, equivalently stated, that there exist at least two distinct elements $x, y \in \{l_\tau, c_\tau, r_\tau\}$ such that $x^\star \vdash x \vdash y$). Now fix $A \coloneqq A_{\zeta, \vdash}$, where

$$(\zeta, \vdash) \in \operatorname*{argmax}_{(\zeta', \vdash') \in \{u, \not u\} \times \{\leq, >\}} \sum_{\tau \in A_{\zeta', \vdash'}} B_\tau \,.$$

In words, $A$ is the set of epochs $A_{\zeta', \vdash'}$ where ReSearch spent the highest budget. Define, for each $\tau \in A$, the points $x_\tau \neq y_\tau$ as the closest and second-closest points in $\{l_\tau, c_\tau, r_\tau\}$ to $x^\star$ such that $x^\star \vdash x_\tau \vdash y_\tau$ (they always exist by definition of $A$). More precisely,

$$x_\tau \coloneqq \operatorname*{argmin}_{x \in \{l_\tau, c_\tau, r_\tau\}, \, x^\star \vdash x} |x - x^\star| \,,$$

$$y_\tau \coloneqq \operatorname*{argmin}_{x \in \{l_\tau, c_\tau, r_\tau\} \setminus \{x_\tau\}, \, x^\star \vdash x} |x - x^\star| \,.$$

Let $n$ be the number of elements of $A$ and $\kappa_1, \ldots, \kappa_n$ be the elements of $A$ in increasing order. Then, for any $i \in \mathbb{N}$, we have

$$|I_{\kappa_i}| \leq |I_{\kappa_{i-1}}|/2 \qquad\qquad \text{if } 2 \leq i \leq n \,, \tag{3}$$

$$|y_{\kappa_i} - x_{\kappa_i}| \leq |y_{\kappa_{i-1}} - x_{\kappa_{i-1}}|/2 \qquad\qquad \text{if } 2 \leq i \leq n \,, \tag{4}$$

$$|x_{\kappa_i} - x^\star| \leq |y_{\kappa_i} - x_{\kappa_i}| \cdot 2 \qquad\qquad \text{if } i \leq n \,, \tag{5}$$

$$|x_{\kappa_i} - x^\star| \leq |x_{\kappa_{i-1}} - x^\star| \qquad\qquad \text{if } 2 \leq i \leq n \,, \tag{6}$$

$$|y_{\kappa_i} - x^\star| \leq |y_{\kappa_{i-1}} - x^\star| \qquad\qquad \text{if } 2 \leq i \leq n \,. \tag{7}$$

Here, equation 3 follows directly by the definition of the update function, noting that there are never two uniform or two non-uniform epochs in a row, unless half of the current interval is eliminated in one single call of the update function. Inequality equation 4 follows directly from equation 3. Inequality equation 5 is a consequence of the definitions of the partition functions $u$ and $\not u$. To prove equation 6, note first that the claim

---

[4]Otherwise, the optimization error is trivially zero.

[5]Indeed, if it does not, then there exists $x^\star \in \{I^-, I^+\}$ such that $\lim_{x \to x^\star, x \in I} f(x) = \inf_{x \in I} f(x)$ (this can only happen if $I$ is not closed or $f$ is discontinuous at $x^\star$; in the latter case, note that by convexity, $f(x^\star) > \lim_{x \to x^\star, x \in I} f(x) = \inf_{x \in I} f(x)$). Thus, noting that ReSearch *never* queries nor recommends the endpoints $\{I^-, I^+\}$ of $I$, one can replace $f$ with $\bar{f}$, where $\bar{f}(x) \coloneqq f(x)$ for all $x \in I$ and $\bar{f}(x^\star) \coloneqq \inf_{x \in I} f(x)$. This way, up to extending (or redefining) $f$ at $x^\star$, we are left with a convex function $\bar{f}$ such that $\bar{f}(X_1) \in J_1, \ldots, \bar{f}(X_T) \in J_T$, attains its minimum at $x^\star$ (in its domain), and satisfies $f(R_T) - \inf_{x \in I} f(x) = \bar{f}(R_T) - \bar{f}(x^\star)$.

holds trivially when $x_{\kappa_{i-1}} \in \{l_{\kappa_i}, c_{\kappa_i}, r_{\kappa_i}\}$. When this is not the case, since ReSearch discarded $x_{\kappa_{i-1}}$ at the end of some previous epoch, it either holds that $I_{\kappa_i} \subset (-\infty, x_{\kappa_{i-1}}]$ or $I_{\kappa_i} \subset [x_{\kappa_{i-1}}, \infty)$. If $x^\star \leq x_{\kappa_{i-1}}$ (meaning that $\vdash = \leq$), it follows from $x^\star \in I_{\kappa_i}$ that $I_{\kappa_i} \subset (-\infty, x_{\kappa_{i-1}}]$, which implies $x^\star \leq x_{\kappa_i} < x_{\kappa_{i-1}}$. Analogously, if $x^\star > x_{\kappa_{i-1}}$ (meaning that $\vdash = >$), it follows from $x^\star \in I_{\kappa_i}$ that $I_{\kappa_i} \subset [x_{\kappa_{i-1}}, \infty)$, which implies $x^\star > x_{\kappa_i} > x_{\kappa_{i-1}}$. This proves equation 6. Moreover, as a direct consequence of equation 4 and equation 6, we obtain equation 7.

By construction, we have that

$$4 \sum_{\tau \in A} B_\tau \geq \sum_{\tau \in [\tau_T]} B_\tau = \sum_{t=1}^{T} b_t \,. \tag{8}$$

Now, we show that for any $\tau \in [\tau_T]$ and $k \in \{0, \ldots, t_\tau\}$, we have

$$\min_{x \in \{l_\tau, c_\tau, r_\tau\}} \mathfrak{B}_{x, t_\tau - k} \geq \frac{B_\tau - (2 + k) \max_{t \in [T]} b_t}{3} \,, \tag{9}$$

i.e., that the total budget $\mathfrak{B}_{x, t_\tau - k}$ spent on any query point $x \in \{l_\tau, c_\tau, r_\tau\}$ up to time $t_\tau - k$ is no smaller (up to $\Theta(k) \cdot \max_{t \in [T]} b_t$) than the budget $B_\tau$ spent (on all query points) only during epoch $\tau$. Indeed, for any $\tau \in [\tau_T]$ and $k \in \{0, \ldots, t_\tau\}$, letting $x_{\min} \in \operatorname{argmin}_{x \in \{l_\tau, c_\tau, r_\tau\}} \mathfrak{B}_{x, t_\tau - k}$ be a query point where the algorithm spent the least amount of budget up to time $t_\tau - k$ and $M \coloneqq \{x \in \{l_\tau, c_\tau, r_\tau\} : \mathfrak{B}_{x, t_\tau - k} - \mathfrak{B}_{x_{\min}, t_\tau - k} \leq \max_{t \in [T]} b_t\}$ be the set of all query points in which the algorithm spent a budget close to that of $x_{\min}$, we have

$$
\begin{aligned}
3 \cdot \min_{x \in \{l_\tau, c_\tau, r_\tau\}} \mathfrak{B}_{x, t_\tau - k} &\geq \sum_{x \in M} \mathfrak{B}_{x, t_\tau - k} - 2 \max_{t \in [T]} b_t \\
&\geq \sum_{x \in M} \sum_{t = t_{\tau-1}+1}^{t_\tau - k} b_t \mathbb{I}\{X_t = x\} - 2 \max_{t \in [T]} b_t \\
&\overset{(*)}{=} B_\tau - \sum_{t = t_\tau - k + 1}^{t_\tau} b_t - 2 \max_{t \in [T]} b_t \\
&\geq B_\tau - (2 + k) \max_{t \in [T]} b_t
\end{aligned}
$$

with the understanding that any sum $\sum_{s=i}^{j} z_s$ is equal to zero whenever $i > j$, and where in $(*)$ we used the fact that, if $x \in \{l_\tau, c_\tau, r_\tau\}$ is such that $\mathfrak{B}_{x, t_\tau - k} - \mathfrak{B}_{x_{\min}, t_\tau - k} > \max_{t \in [T]} b_t$, then ReSearch never queried $x$ in epoch $\tau$ up to time $t_\tau - k$, i.e.,

$$\sum_{t = t_{\tau-1}+1}^{t_\tau - k} b_t = 0.$$

This proves equation 9.

Thus, for any $\tau \in A$, if $B_\tau > 3 \max_{t \in [T]} b_t$, since $J_{x_\tau, t_\tau - 1}^{+} > J_{y_\tau, t_\tau - 1}^{-}$ (this follows from the definition of the delete function and can be proved by exhaustion), it holds that

$$
\begin{aligned}
f(y_\tau) - f(x_\tau) &\leq J_{y_\tau, t_\tau - 1}^{+} - J_{x_\tau, t_\tau - 1}^{-} \\
&< J_{y_\tau, t_\tau - 1}^{+} - J_{y_\tau, t_\tau - 1}^{-} + J_{x_\tau, t_\tau - 1}^{+} - J_{x_\tau, t_\tau - 1}^{-} \\
&= |J_{y_\tau, t_\tau - 1}| + |J_{x_\tau, t_\tau - 1}| \leq \frac{c}{\mathfrak{B}_{y_\tau, t_\tau - 1}^{\alpha}} + \frac{c}{\mathfrak{B}_{x_\tau, t_\tau - 1}^{\alpha}} \\
&\leq \frac{2 \cdot c}{\left(\min_{x \in \{l_\tau, c_\tau, r_\tau\}} \mathfrak{B}_{x, t_\tau - 1}\right)^{\alpha}} \\
&\overset{\text{equation 9}}{\leq} \frac{3^{\alpha} \cdot 2 \cdot c}{\left(B_\tau - 3 \max_{t \in [T]} b_t\right)^{\alpha}} \,. \tag{10}
\end{aligned}
$$

Assume now that $f(y_{\kappa_n}) - f(x_{\kappa_n}) > 0$. Then, for any $\tau \in A$, by convexity and inequalities equation 6–equation 7, we have that $f(y_\tau) - f(x_\tau) > 0$ is also true. By equation 10, it follows that, for any $i \in [n]$,

regardless of the fact that the inequality $B_{\kappa_i} > 3 \max_{t \in [T]} b_t$ holds or not,

$$B_{\kappa_i} \leq 3 \max_{t \in [T]} b_t + \frac{3 \cdot (2 \cdot c)^{1/\alpha}}{\left(f(y_{\kappa_i}) - f(x_{\kappa_i})\right)^{1/\alpha}} \cdot \tag{11}$$

Summing equation 11 over $i \in [a]$, we obtain

$$\sum_{t=1}^{T} b_t \overset{\text{equation 8}}{\leq} 4 \sum_{\tau \in A} B_\tau = 4 \sum_{i=1}^{n} B_{\kappa_i}$$

$$\overset{\text{equation 11}}{\leq} 4 \sum_{i=1}^{n} \left(3 \max_{t \in [T]} b_t + \frac{3 \cdot (2 \cdot c)^{1/\alpha}}{\left(f(y_{\kappa_i}) - f(x_{\kappa_i})\right)^{1/\alpha}}\right)$$

$$= 12 \cdot \max_{t \in [T]} b_t \cdot n + \sum_{i=1}^{n} \frac{12 \cdot (2 \cdot c)^{1/\alpha}}{\left(f(y_{\kappa_i}) - f(x_{\kappa_i})\right)^{1/\alpha}} \cdot \tag{12}$$

Now, using equation 6–equation 7 together with the fact that difference quotients of a convex function are non-decreasing in both variables, and since for each $i \in [n]$ it holds that $(y_{\kappa_n} - x_{\kappa_n}) \cdot (y_{\kappa_i} - x_{\kappa_i}) > 0$, we have

$$\sum_{i=1}^{n} \frac{\left(f(y_{\kappa_n}) - f(x_{\kappa_n})\right)^{1/\alpha}}{\left(f(y_{\kappa_i}) - f(x_{\kappa_i})\right)^{1/\alpha}} = \sum_{i=1}^{n} \left(\frac{f(y_{\kappa_n}) - f(x_{\kappa_n})}{f(y_{\kappa_i}) - f(x_{\kappa_i})}\right)^{1/\alpha}$$

$$= \sum_{i=1}^{n} \left(\left(\frac{\frac{f(y_{\kappa_n}) - f(x_{\kappa_n})}{y_{\kappa_n} - x_{\kappa_n}}}{\frac{f(y_{\kappa_i}) - f(x_{\kappa_i})}{y_{\kappa_i} - x_{\kappa_i}}}\right)^{1/\alpha} \cdot \left|\frac{y_{\kappa_n} - x_{\kappa_n}}{y_{\kappa_i} - x_{\kappa_i}}\right|^{1/\alpha}\right)$$

$$\overset{\text{equation 6+equation 7}}{\leq} \sum_{i=1}^{n} \left|\frac{y_{\kappa_n} - x_{\kappa_n}}{y_{\kappa_i} - x_{\kappa_i}}\right|^{1/\alpha} \overset{\text{equation 4}}{\leq} \sum_{i=1}^{n} \left(\frac{1}{2^{n-i}}\right)^{1/\alpha} \leq \frac{1}{1 - 2^{-1/\alpha}} \cdot \tag{13}$$

Putting equation 12 and equation 13 together, we obtain the inequality

$$\sum_{t=1}^{T} b_t \leq 12 \cdot \max_{t \in [T]} b_t \cdot n + \frac{12 \cdot (2 \cdot c)^{1/\alpha} \cdot \frac{1}{1 - 2^{-1/\alpha}}}{\left(f(y_{\kappa_n}) - f(x_{\kappa_n})\right)^{1/\alpha}} \,,$$

that can be rearranged, whenever $\sum_{t=1}^{T} b_t \geq 24 \cdot \max_{t \in [T]} b_t \cdot n$, to obtain the inequality

$$f(y_{\kappa_n}) - f(x_{\kappa_n}) \leq 4 \cdot \left(\frac{24}{2^{1/\alpha} - 1}\right)^{\alpha} \cdot \frac{c}{(\sum_{t=1}^{T} b_t)^{\alpha}} \cdot \tag{14}$$

Then, relying again on the fact that difference quotients of a convex function are non-decreasing in both variables, and that $(y_{k_n} - x_{k_n}) \cdot (x_{k_n} - x^\star) \geq 0$, whenever $\sum_{t=1}^{T} b_t \geq 24 \cdot \max_{t \in [T]} b_t \cdot n$ and $x_{\kappa_n} \neq x^\star$, we have that

$$f(x_{\kappa_n}) - f(x^\star) = \frac{f(x_{\kappa_n}) - f(x^\star)}{x_{\kappa_n} - x^\star} \cdot (x_{\kappa_n} - x^\star)$$

$$\leq \frac{f(y_{\kappa_n}) - f(x_{\kappa_n})}{y_{\kappa_n} - x_{\kappa_n}} \cdot (x_{\kappa_n} - x^\star)$$

$$\overset{\text{equation 14}}{\leq} 4 \cdot \left(\frac{24}{2^{1/\alpha} - 1}\right)^{\alpha} \cdot \frac{c}{(\sum_{t=1}^{T} b_t)^{\alpha}} \cdot \left|\frac{x_{\kappa_n} - x^\star}{y_{\kappa_n} - x_{\kappa_n}}\right|$$

$$\overset{\text{equation 5}}{\leq} 8 \cdot \left(\frac{24}{2^{1/\alpha} - 1}\right)^{\alpha} \cdot \frac{c}{(\sum_{t=1}^{T} b_t)^{\alpha}} \tag{15}$$

(note that equation 15 is trivially true also when $x_{\kappa_n} = x^\star$) and

$$f(y_{\kappa_n}) - f(x^\star) = f(y_{\kappa_n}) - f(x_{\kappa_n}) + f(x_{\kappa_n}) - f(x^\star)$$

$$\overset{\text{equation 14+equation 15}}{\leq} 12 \cdot \left(\frac{24}{2^{1/\alpha} - 1}\right)^\alpha \cdot \frac{c}{(\sum_{t=1}^T b_t)^\alpha} \ . \tag{16}$$

Recall that the derivations for equation 15 and equation 16 were carried out under the assumption that $f(y_{\kappa_n}) - f(x_{\kappa_n}) > 0$. If this assumption does not hold, by convexity, $f(y_{\kappa_n}) = f(x_{\kappa_n}) = f(x^\star)$, therefore equation 15 and equation 16 are still (trivially) true.

Now, we will prove that the recommendation $R_T$ is approximately at least as good as $x_{\kappa_n}$ and $y_{\kappa_n}$. Since under the assumption that $\sum_{t=1}^T b_t \geq 24 \cdot n \cdot \max_{t \in [T]} b_t$ we showed in equation 15 and equation 16 that $x_{\kappa_n}$ and $y_{\kappa_n}$ are both near-minimizers, this will yield under the same assumption that $R_T$ is also a near-minimizer.

Recalling that we are currently assuming $\text{del}_T \neq \square\square\square\square$, we have that $R_T \in \text{argmin}_{x \in \{l_{\tau_T+1}, c_{\tau_T+1}, r_{\tau_T+1}\}} J_{x,T}^+$, which (as can be checked directly) implies in turn that $R_T \in \text{argmin}_{x \in \{l_{\tau_T}, c_{\tau_T}, r_{\tau_T}\}} J_{x,T}^+$ and, whenever $\sum_{t=1}^T b_t \geq 24 \cdot n \cdot \max_{t \in [T]} b_t$:

1. If $R_T \in \{x_{\kappa_n}, y_{\kappa_n}\}$, then equation 1 follows by equation 15 and equation 16.

2. If $R_T \notin \{x_{\kappa_n}, y_{\kappa_n}\}$, and $\{x_{\kappa_n}, y_{\kappa_n}\} \subset \{l_{\tau_T}, c_{\tau_T}, r_{\tau_T}\}$, then there exists $x \in \{l_{\tau_T}, c_{\tau_T}, r_{\tau_T}\} \setminus \{R_T\} = \{x_{\kappa_n}, y_{\kappa_n}\}$ such that $f(R_T) \leq J_{R_T,T}^+ \leq J_{x,T}^- \leq f(x)$; therefore, equation 1 follows by equation 15 and equation 16.

3. If $R_T \notin \{x_{\kappa_n}, y_{\kappa_n}\}$, and $\{x_{\kappa_n}, y_{\kappa_n}\} \not\subset \{l_{\tau_T}, c_{\tau_T}, r_{\tau_T}\}$, then, since at least one between $x_{\kappa_n}$ and $y_{\kappa_n}$ does not belong to the set of active query points $\{l_{\tau_T}, c_{\tau_T}, r_{\tau_T}\}$ at time $T$, there exist a past time $t \in [T]$ and a past query point $x \in \{l_{\tau_t}, c_{\tau_t}, r_{\tau_t}\}$ such that $J_{x,t}^+ \leq \max\big(J_{x_{\kappa_n},t}^-, J_{y_{\kappa_n},t}^-\big)$; therefore, noting that the sequence $s \mapsto \min_{x' \in \{l_{\tau_s}, c_{\tau_s}, r_{\tau_s}\}} J_{x',s}^+$ is non-increasing, we have $f(R_T) \leq J_{R_T,T}^+ \leq J_{x,t}^+ \leq \max\big(J_{x_{\kappa_n},t}^-, J_{y_{\kappa_n},t}^-\big) \leq \max\big(f(x_{\kappa_n}), f(y_{\kappa_n})\big)$ and equation 1 follows by equation 15 and equation 16.

On the other hand, if $\sum_{t=1}^T b_t < 24 \cdot n \cdot \max_{t \in [T]} b_t$, then

$$f(R_T) - f(x^\star) \leq \frac{3}{4} L \left|I_{\tau_T+1}\right| \leq \frac{9}{16} L \left|I_{\kappa_n}\right|$$

$$\overset{\text{equation 3}}{\leq} \frac{9}{16} L |I| (1/2)^{n-1} \leq \frac{9}{16} L |I| (1/2)^{\frac{\sum_{t=1}^T b_t}{24 \max_{t \in [T]} b_t} - 1}$$

$$= \frac{9}{8} L |I| \exp\left(-\frac{\ln 2}{24} \frac{\sum_{t=1}^T b_t}{\max_{t \in [T]} b_t}\right),$$

where $L$ is the smallest between the Lipschitz constants of $f$ on $[l_{\tau_T}, r_{\tau_T}]$ and $[l_{\tau_T+1}, r_{\tau_T+1}]$ —indeed, by convexity, $L$ is a Lipschitz constant for $f$ on the convex hull of $\{x^\star, l_{\tau_T}, r_{\tau_T}\}$ (resp., $\{x^\star, l_{\tau_T+1}, r_{\tau_T+1}\}$) if and only if it is a Lipschitz constant on $[l_{\tau_T}, r_{\tau_T}]$ (resp., $[l_{\tau_T+1}, r_{\tau_T+1}]$). Putting everything together yields equation 1 when $\text{del}_T \neq \square\square\square\square$.

Assume now that $\text{del}_T = \square\square\square\square$ and $B_{\tau_T} \geq \sum_{\tau'=0}^{\tau_T-1} B_{\tau'}$ (i.e., the condition on Line 12 is true and we recommend $R_T$ as in Line 13). This implies that the three intervals $J_{l_{\tau_T},T}, J_{c_{\tau_T},T}, J_{r_{\tau_T},T}$ have non-empty intersection, which in turn implies that

$$\max_{x' \in \{l_{\tau_T}, c_{\tau_T}, r_{\tau_T}\}} J_{x',T}^+ - \min_{x' \in \{l_{\tau_T}, c_{\tau_T}, r_{\tau_T}\}} J_{x',T}^- \leq 2 \max_{x' \in \{l_{\tau_T}, c_{\tau_T}, r_{\tau_T}\}} \left|J_{x',T}\right| \ . \tag{17}$$

Now, we define $f_1, f_2, f_3, f_4$ as the four functions whose graphs are straight lines such that $f_1$ passes through the points $(l_{\tau_T}, J_{l_{\tau_T},T}^-)$ and $(r_{\tau_T}, J_{r_{\tau_T},T}^-)$, $f_2$ passes through the points $(c_{\tau_T}, J_{c_{\tau_T},T}^+)$ and $(r_{\tau_T}, J_{r_{\tau_T},T}^+)$, $f_3$ passes through the points $(l_{\tau_T}, J_{l_{\tau_T},T}^+)$ and $(c_{\tau_T}, J_{c_{\tau_T},T}^-)$, and $f_4$ passes through the points $(l_{\tau_T}, J_{l_{\tau_T},T}^+)$ and $(r_{\tau_T}, J_{r_{\tau_T},T}^-)$ (Figure 5).

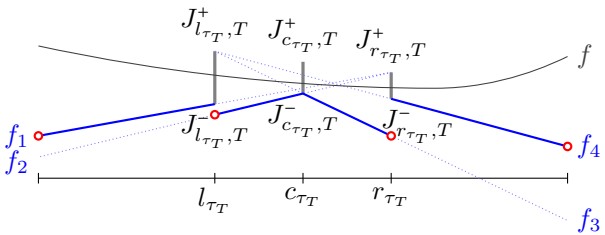

Figure 5: A representation of the four lines $f_1, \ldots, f_4$. By convexity, $f$ is lower bounded by the blue solid segments. Note that, since $J_{l_{\tau_T},T} \cap J_{c_{\tau_T},T} \cap J_{r_{\tau_T},T} \neq \varnothing$, then $f_1, f_2$ are nondecreasing and $f_3, f_4$ nonincreasing. Therefore, the minimum of the $y$ coordinates of the red dots is a lower bound on the value of the function, which in turn implies equation 18.

By the convexity of $f$, for each $x \in [I_{\tau_T}^-, l_{\tau_T}]$ we have $f(x) \geq f_1(x)$, for each $x \in [l_{\tau_T}, c_{\tau_T}]$ we have $f(x) \geq f_2(x)$, for each $x \in [c_{\tau_T}, r_{\tau_T}]$ we have $f(x) \geq f_3(x)$, and for each $x \in [r_{\tau_T}, I_{\tau_T}^+]$ we have $f(x) \geq f_4(x)$. Writing down explicitly these four inequalities and upper bounding, we conclude that

$$
\begin{aligned}
f(x^\star) \quad &\geq \min_{x' \in \{l_{\tau_T}, c_{\tau_T}, r_{\tau_T}\}} J_{x,T}^- \\
&- 2 \left( \max_{x' \in \{l_{\tau_T}, c_{\tau_T}, r_{\tau_T}\}} J_{x',T}^+ - \min_{x' \in \{l_{\tau_T}, c_{\tau_T}, r_{\tau_T}\}} J_{x',T}^- \right) .
\end{aligned}
\tag{18}
$$

Then, if $B_{\tau_T} \geq 4 \max_{t \in [T]} b_t$, for all $x \in \{l_{\tau_T}, c_{\tau_T}, r_{\tau_T}\}$, we have

$$
\begin{aligned}
f(x) - f(x^\star) &\leq J_{x,T}^+ - f(x^\star) \\
&\overset{\text{equation } 18}{\leq} J_{x,T}^+ - \min_{x' \in \{l_{\tau_T}, c_{\tau_T}, r_{\tau_T}\}} J_{x,T}^- \\
&\quad + 2 \left( \max_{x' \in \{l_{\tau_T}, c_{\tau_T}, r_{\tau_T}\}} J_{x',T}^+ - \min_{x' \in \{l_{\tau_T}, c_{\tau_T}, r_{\tau_T}\}} J_{x',T}^- \right) \\
&\overset{\text{equation } 17}{\leq} 6 \max_{x' \in \{l_{\tau_T}, c_{\tau_T}, r_{\tau_T}\}} |J_{x',T}| \leq 6 \max_{x' \in \{l_{\tau_T}, c_{\tau_T}, r_{\tau_T}\}} \frac{c}{\mathfrak{B}_{x',T}^{\alpha}} \\
&\overset{\text{equation } 9}{\leq} 6 \frac{c}{\left( \frac{B_{\tau_T} - 2 \max_{t \in [T]} b_t}{3} \right)^{\alpha}} \leq 6 \cdot 3^{\alpha} \frac{c}{(B_{\tau_T}/2)^{\alpha}} \\
&\leq 6 \cdot 12^{\alpha} \cdot \frac{c}{(\sum_{t=1}^{T} b_t)^{\alpha}} \leq 12 \cdot \left( \frac{48}{2^{1/\alpha} - 1} \right)^{\alpha} \cdot \frac{c}{(\sum_{t=1}^{T} b_t)^{\alpha}} .
\end{aligned}
$$

If, on the other hand, $B_{\tau_T} < 4 \max_{t \in [T]} b_t$, since $B_{\tau_T} \geq \frac{1}{2} \sum_{t=1}^{T} b_t$, then for all $x \in \{l_{\tau_T}, c_{\tau_T}, r_{\tau_T}\}$, we have

$$
\begin{aligned}
f(x) - f(x^\star) &\leq L |I| (3/4) \leq L |I| (1/2)^{\frac{4 \max_{t \in [T]} b_t}{12 \max_{t \in [T]} b_t}} \\
&\leq L |I| (1/2)^{\frac{B_{\tau_T}}{12 \max_{t \in [T]} b_t}} \leq L |I| (1/2)^{\frac{\sum_{t=1}^{T} b_t}{24 \max_{t \in [T]} b_t}} \\
&= L |I| \exp \left( -\frac{\ln 2}{24} \frac{\sum_{t=1}^{T} b_t}{\max_{t \in [T]} b_t} \right)
\end{aligned}
$$

where $L$ the Lipschitz constant of $f$ on $[l_{\tau_T}, r_{\tau_T}]$. Thus, adding together the two bounds for $B_{\tau_T} \geq 4 \max_{t \in [T]} b_t$ and $B_{\tau_T} < 4 \max_{t \in [T]} b_t$ yields equation 1 when $\mathrm{del}_T = \square\square\square\square$ and $B_{\tau_T} \geq \sum_{\tau'=0}^{\tau_T - 1} B_{\tau'}$.

Finally, assume that $\mathrm{del}_T = \square\square\square\square$ and $B_{\tau_T} < \sum_{\tau'=0}^{\tau_T - 1} B_{\tau'}$ (i.e., the condition on Line 14 is true and we recommend $R_T$ as in Line 15). Then the recommendation $R_T$ is the point with the best upper bound at the

end of the second-to-last epoch. Thus, proceeding as in the first part of the proof (case $\text{del}_T \neq \square\square\square\square$), we get

$$
\begin{aligned}
f(R_T) - f(x^\star) &= f\left(R_{t_{\tau_{T}-1}}\right) - f(x^\star) \\
&\leq 12\left(\frac{24}{2^{1/\alpha}-1}\right)^\alpha \frac{c}{(\sum_{t=1}^{t_{\tau_T-1}} b_t)^\alpha} \\
&\quad + \frac{9}{8} L\,|I| \exp\left(-\frac{\ln 2}{24} \frac{\sum_{t=1}^{t_{\tau_T-1}} b_t}{\max_{t \in [t_{\tau_T-1}]} b_t}\right) \\
&< 12\left(\frac{48}{2^{1/\alpha}-1}\right)^\alpha \frac{c}{(\sum_{t=1}^{T} b_t)^\alpha} \\
&\quad + \frac{9}{8} L\,|I| \exp\left(-\frac{\ln 2}{48} \frac{\sum_{t=1}^{T} b_t}{\max_{t \in [T]} b_t}\right)
\end{aligned}
$$

where $L$ the Lipschitz constant of $f$ on $[l_{\tau_T}, r_{\tau_T}]$. Being the interval $I$, the time $T$ and the convex function $f$ arbitrarily chosen, the proof is complete. $\qquad\square$

## B  Full proof of Theorem 3.2

In this section, we give a detailed proof of Theorem 3.2.

*Proof.* Fix a nondegenerate bounded interval $I \subset \mathbb{R}$. Fix also an horizon $T \in \mathbb{N}$, a total budget $B > 0$, and a Lipschitz constant $L > 0$. For each $t \in [T]$, define $b_t := B/T$. We divide the proof in two cases, depending on which of the two addends in equation 2 is the dominant term.

Assume first $\frac{1}{4} \cdot \frac{c}{(\sum_{t=1}^{T} b_t)^\alpha} \geq \frac{1}{32e} \cdot L\,|I| \exp\left(-\frac{\sum_{t=1}^{T} b_t}{\max_{t \in [T]} b_t}\right)$. For all $b > 0$, define $J(b) := \left[-c/(2b^\alpha), c/(2b^\alpha)\right]$. Consider the two alternative objective functions $f_+$ and $f_-$, defined for all $x \in I$, by

$$
f_\pm(x) := \pm\left(1 - \frac{2(x - I^-)}{|I|}\right) \cdot \frac{c}{2B^\alpha}\,.
$$

At each time $t \in [T]$, if the algorithm chosen by the optimizer queried $X_1, \ldots, X_t$, the environment returns the fuzzy evaluation $J_t := J(\mathfrak{B}_{X_t,t})$, where we recall that $\mathfrak{B}_{x,t}$ was defined, for any $x \in I$, by $\sum_{s=1}^{t} b_s \mathbb{I}\{X_s = x\}$. Note that the environment satisfies Assumption 2.1 and that both functions $f_\pm$ are $\frac{c}{|I|B^\alpha}$-Lipschitz. Moreover, the algorithm provides the same queries and recommendations for both $f^-$ and $f^+$, as it receives the same $J_1, \ldots, J_T$. Furthermore, if the algorithm recommends $R_T \geq (I^- + I^+)/2$ then $f_-(R_T) - \inf_{x \in I} f_-(x) \geq c/(2B^\alpha)$, while if the algorithm recommends $R_T < (I^- + I^+)/2$ then $f_+(R_T) - \inf_{x \in I} f_+(x) \geq c/(2B^\alpha)$. Thus, in both cases there exists a $\frac{c}{|I|B^\alpha}$-Lipschitz convex function $f \in \{f_-, f_+\}$ for which:

$$
f(R_T) - \inf_{x \in I} f(x) \geq \frac{1}{4} \cdot \frac{c}{(\sum_{t=1}^{T} b_t)^\alpha} + \frac{1}{32e} L\,|I|\, e^{-\frac{\sum_{t=1}^{T} b_t}{\max_{t \in [T]} b_t}}\,.
$$

Assume now $\frac{1}{4} \cdot \frac{c}{(\sum_{t=1}^{T} b_t)^\alpha} < \frac{1}{32e} \cdot L\,|I| \exp\left(-\frac{\sum_{t=1}^{T} b_t}{\max_{t \in [T]} b_t}\right)$. In this case, at each time $t \in [T]$, the environment returns $J_t := \{f(X_t)\}$. Note that, in this instance, our problem reduces to *deterministic* convex optimization. By a classic lower bound for deterministic convex optimization (see, e.g., Nesterov et al. 2018, Theorem 3.2.8), then, there exists an $L$-Lipschitz convex function $f : I \to \mathbb{R}$ for which

$$
\begin{aligned}
f(R_T) - \inf_{x \in I} f(x) &> \frac{1}{16e} L\,|I|\, e^{-T} = \frac{1}{16e} L\,|I|\, e^{-\frac{\sum_{t=1}^{T} b_t}{\max_{t \in [T]} b_t}} \\
&\geq \frac{1}{4} \cdot \frac{c}{(\sum_{t=1}^{T} b_t)^\alpha} + \frac{1}{32e} L\,|I|\, e^{-\frac{\sum_{t=1}^{T} b_t}{\max_{t \in [T]} b_t}}\,.
\end{aligned}
$$

Being the interval $I$, the horizon $T$, the budget $B$, and the Lipschitz constant $L$ arbitrarily chosen, the theorem follows. $\qquad\square$

## C   Full proof of Theorem 4.2

In this section, we give a detailed proof of Theorem 4.2.

*Proof.* For the sake of simplicity, we assume that $I \coloneqq [0,1]$, $f$ is continuous and admits a unique minimizer $x^\star \in [0,1]$ (the other cases can be treated similarly). For each $n \in \mathbb{N}$, let $\mathcal{D}_n \coloneqq \{k \cdot 2^{-n} \mid k \in \mathbb{Z}\}$, let $\mathcal{D}_n^\star \coloneqq \{x_{n,1}, \ldots, x_{n,10}\} \subset \mathcal{D}_n$ such that

$$x_{n,1} < \cdots < x_{n,5} \le x^\star \le x_{n,6} < \cdots < x_{n,10}$$

and $x_{n,j+1} - x_{n,j} \le 2^{-n}$, for all $j \in [9]$. Define $\mathcal{D} \coloneqq \bigcup_{n\in[T]} \mathcal{D}_n^\star \cap (0,1)$. Consider the "good event"

$$\mathcal{E} \coloneqq \bigcap_{\substack{n,t\in[T] \\ j\in[10]}} \left\{ \left| m(Y_{x_{n,j},1}, \ldots, Y_{x_{n,j},t}) - f(x_{n,j}) \right| \le \frac{c(\delta)}{t^\alpha} \right\}$$

and note that $\bigcap_{t\in[T]}\{f(X_t) \in J_t\} \subset \mathcal{E}$. By De Morgan's laws, a union bound, and Assumption 4.1, we have $\mathbb{P}[\mathcal{E}^c] \le 10T^2\delta$. Now, if we are in the good event $\mathcal{E}$, then Assumption 2.1 holds for all $t \in [T]$, with $c = c(\delta)$ and $b_1 = \cdots = b_T = 1$, since ReSearch queries points only in $\mathcal{D}$. To prove the last claim, consider the budget version of ReSearch. Recall the observation (at the beginning of the proof in Appendix A) that the minimizer $x^\star$ of $f$ belongs to all active intervals of ReSearch at all epochs. Assume by contradiction that there exists an epoch $\tau \in [\tau_T]$ such that $\{l_\tau, c_\tau, r_\tau\}$ is not included in $\mathcal{D}$. Then, the set of query points $\{l_\tau, c_\tau, r_\tau\}$ is not included in $\mathcal{D}_n^\star \cap (0,1)$ when $n = -\log_2\big((r_\tau - l_\tau)/2\big)$. Consider the case where $r_\tau \notin \mathcal{D}_n^\star \cap (0,1)$ and $r_\tau > x_{n,10}$ (the other cases can be analyzed analogously). Since the leftmost point $I_\tau^-$ of the active interval of epoch $\tau$ is always bigger than or equal to $r_\tau - 4 \cdot 2^{-n}$, then

$$x^\star \ge I_\tau^- \ge r_\tau - 4 \cdot 2^{-n} \ge x_{n,10} + 2^{-n} - 4 \cdot 2^{-n} = x_{n,7} > x_{n,6} \ge x^\star$$

which yields a contradiction. It follows that, in the good event $\mathcal{E}$ (hence, with probability at least $1 - 10T^2\delta$), we can apply Theorem 3.1, obtaining the result. $\qquad\square$

## D   Missing details on Section 4.2

We now consider the minimization of a multivariate objective over a convex bounded subset $I \subset \mathbb{R}^d$, where Optimization Protocol 1 has $b_t = 1$ for each $t$ and Assumption 2.1 holds. In this section, we refer to this setting as Multivariate Budget Convex Optimization (MBCO) for the sake of emphasis. We assume the objective to be $\mu$ strongly convex and $\beta$-smooth with $0 < \mu \le \beta$ i.e., $\forall x, x_0 \in \mathbb{R}^d$

$$f(x) \ge f(x_0) + \langle \nabla f(x_0), x - x_0 \rangle + \frac{\mu}{2}\|x - x_0\|^2$$
$$f(x) \le f(x_0) + \langle \nabla f(x_0), x - x_0 \rangle + \frac{\beta}{2}\|x - x_0\|^2$$

where the first equation corresponds to the strong convexity and the second to the smoothness. For simplicity, we further assume the existence of a unique minimizer[6] $x^* \in \text{int}(I)$. As it will be apparent later, it is not necessary to assume that $x^*$ is in the interior of $I$: the entire analysis that follows holds if the objective is Lipschitz, a condition that is implied by the strong convexity and smoothness. This cosmetic choice allows us to replace the local Lipschitz constant appearing in equation 1 with $\beta$.

Before providing the detailed version of Theorem 4.3, let us provide some intuition on the bounds appearing there. Given some budget $\bar{T}$ for a line search subroutine, the bound of equation 1 scales roughly as $1/\bar{T}^\alpha$, which implies that the point recommended by ReSearch is at distance at most $1/\bar{T}^{\alpha/2}$ from $x^*$ due to strong convexity. We denote with $T(\eta) = \lceil \left(\frac{4c_1}{\mu\eta^2}\right)^{\frac{1}{\alpha}} \rceil$ the maximum number of iterations needed by ReSearch to find

---

[6]The existence of a unique minimizer is an easy consequence of the strong convexity of $f$.

an $\eta$-minimizer in a given epoch and by $\bar{K}(T,\eta) \coloneqq \lfloor T/T(\eta) \rfloor$ the corresponding minimum number of epochs made by the algorithm with an overall budget of $T$ (we have approximately $K(T,\eta) \geq \bar{K}(T,\eta)$ with the notation of Theorem 4.3).

Now, we state the detailed version of Theorem 4.3.

**Theorem D.1.** *Let $\Delta_0 \coloneqq f(x_0) - f(x^*)$, $x^*$ be the unique minimizer, $\kappa \coloneqq \beta/\mu$ be the condition number of $f$, and $\Theta \coloneqq \max_{i \in [d]} |I_k|$. Recall $c_1, c_2, c_3$ from Theorem 3.1. Suppose that $\alpha \geq \frac{\ln 2}{48}$ and*

$$T \geq \max\left\{ (2d\kappa\beta)^{\frac{1}{\alpha}}, \frac{1}{\log\left(1 - \frac{1}{4d\kappa}\right)}, 2\frac{\alpha}{c_3}\ln\left(\frac{\alpha}{c_3}\right), 4\frac{\alpha}{c_3}\ln\left(2\frac{\alpha}{c_3}\right) + \frac{2}{c_3}\ln\left(\frac{c_2\beta\Theta\mathrm{diam}(I)}{c_1}\right) \right\}.$$

*If the optimizer runs ReSearch for MBCO (Algorithm 5), then, its expected optimization error is upper bounded by*

$$\left(1 - \frac{1}{4d\kappa}\right)^{\bar{K}(T,\eta)} \Delta_0 + 2d\kappa\beta\eta^2. \tag{19}$$

*Thus by setting $\eta = \left(\frac{1}{c(\alpha)T\log\left(1 - \frac{1}{4d\kappa}\right)}\log\left(\frac{2d\kappa\beta}{T^\alpha}\right)\right)^{\frac{\alpha}{2}}$ with $c(\alpha) = (\mu/(4c_1))^{1/\alpha}$, the expected optimization error is upper bounded by*

$$\frac{2d\kappa\beta}{T^\alpha}\left(\frac{\Delta_0}{\left(1 - \frac{1}{4d\kappa}\right)} + \left(\frac{1}{c(\alpha)\log\left(1 - \frac{1}{4d\kappa}\right)}\log\left(\frac{2d\kappa\beta}{T^\alpha}\right)\right)^\alpha\right). \tag{20}$$

*Proof of Theorem D.1.* Take $\alpha, \mu, \beta, \kappa$ and $\Theta$ as defined in Theorem D.1. Let $g$ be a univariate function obtained by considering $f$ on a segment $I'$ of $I$. We can upper bound the Lipschitz constant of the function $g$ using the global Lipschitz constant $L$ of $f$, which, given that we are assuming that $x^* \in I^\circ$, can be further upper bounded by $\beta \cdot \mathrm{diam}(I)$. We start noticing that, assuming $b_t = 1$ for all $t$ and $c = 1$, the bound of equation 1 can be bounded from above by $2c_1/T^\alpha$ whenever

$$T \geq \max\left\{2\frac{\alpha}{c_3}\ln\left(\frac{\alpha}{c_3}\right), 4\frac{\alpha}{c_3}\ln\left(2\frac{\alpha}{c_3}\right) + \frac{2}{c_3}\ln\left(\frac{c_2 L\Theta}{c_1}\right)\right\}. \tag{21}$$

Indeed,

$$\frac{c_1}{T^\alpha} \geq c_2 L\Theta \exp\left(-c_3 T\right), \tag{22}$$

is equivalent to

$$\frac{c_1}{c_2 L\Theta} \geq T^\alpha \exp\left(-c_3 T\right).$$

Taking logs both sides and re-arranging this is equivalent to

$$T \geq \frac{\alpha}{c_3}\ln(T) + \frac{1}{c_3}\ln\left(\frac{c_2 L\Theta}{c_1}\right).$$

Now consider the case when $c_2 L\Theta \geq c_1$, so that the second term on the right hand side is positive. Notice that by hypothesis we have that $\alpha/c_3 \geq 1$, so by Lemma A.2 from Appendix A of Shalev-Shwartz & Ben-David (2014), by taking

$$T \geq 4\frac{\alpha}{c_3}\ln\left(2\frac{\alpha}{c_3}\right) + \frac{2}{c_3}\ln\left(\frac{c_2 L\Theta}{c_1}\right),$$

equation 22 holds. On the other hand, if $c_2 L\Theta < c_1$, then the second term on the right hand side is negative and we can solve the stronger inequality

$$T \geq \frac{\alpha}{c_3} \ln(T),$$

using Lemma A.1 from Appendix A of Shalev-Shwartz & Ben-David (2014). Thus by taking

$$T \geq 2\frac{\alpha}{c_3} \ln\left(\frac{\alpha}{c_3}\right),$$

equation 22 holds.

Now let $T$ any integer satisfying equation 21 and let denote with $x_T$ the output of ReSearch after $T$ iterations. Notice that for any function $g$ as described above, by the strong convexity of $g$ it holds that

$$|x_T - x^*| \leq \sqrt{\frac{2}{\mu}(g(x_T) - g^*)} \leq 2\sqrt{\frac{c_1}{\mu T^\alpha}}.$$

Let $0 < \eta \leq \Theta$, if $T(\eta) = \lceil \left(\frac{4c_1}{\mu\eta^2}\right)^{\frac{1}{\alpha}} \rceil$, then it follows that the point $x_T$ found by ReSearch satisfies $|x_T - x^*| \leq \eta$. Now let $f$ be a multi-variate function satisfying the assumption of Theorem D.1. Notice that any restriction $f_k$ of $f$ along a coordinate line $k$, will also satisfies the same assumptions over the interval $I_k$ (see line 3 of Algorithm 5).

The analysis of Jamieson et al. (2012) applies to our case and they show that after $k$ epochs of coordinate descend, the expected optimization error of the current iterate is bounded above by

$$\left(1 - \frac{1}{4d\kappa}\right)^k \Delta_0 + 2d\kappa\beta\eta^2.$$

Recalling that $T(\eta)$ is the worst-case number of iterations required by ReSearch to find an $\eta$-optimizer, and if we are given a total budget of $T$, the number of epochs made by equation 5 is at least $\bar{K}(T, \eta) = \lfloor T/T(\eta) \rfloor$. From this we get the bound of equation 19. Now we set

$$\eta = \left(\frac{2}{c(\alpha)T\log\left(1 - \frac{1}{4d\kappa}\right)} \log\left(\frac{2d\kappa\beta}{T^\alpha}\right)\right)^{\frac{\alpha}{2}},$$

whit $c(\alpha) = (\mu/(4c_1))^{1/\alpha}$, and notice that if $T > \max\{1/(\log(1 - 1/(4d\kappa))), (2d\kappa\beta)^{1/\alpha}\}$, then $T(\eta) \leq 2(4c_1/(\mu\eta^2))^{1/\alpha} > 1/2$. Thus, the following holds

$$\bar{K}(T, \eta) = \lfloor T/T(\eta) \rfloor \geq T/T(\eta) - 1 \geq \frac{T}{2\left(\frac{4c_1}{\mu\eta^2}\right)^{\frac{1}{\alpha}}} - 1 = \frac{T}{2}\left(\frac{\mu}{4c_1}\right)^{\frac{1}{\alpha}} \eta^{\frac{2}{\alpha}} - 1$$

$$= \left(\frac{1}{\log\left(1 - \frac{1}{4d\kappa}\right)} \log\left(\frac{2d\kappa\beta}{T^\alpha}\right)\right) - 1.$$

Replacing this into equation 19

$$\left(1 - \frac{1}{4d\kappa}\right)^{\bar{K}(T,\eta)} \Delta_0 + 2d\kappa\beta\eta^2$$

$$\leq \frac{\frac{2d\kappa\beta}{T^\alpha}}{\left(1 - \frac{1}{4d\kappa}\right)}\Delta_0 + 2d\kappa\beta\left(\frac{1}{c(\alpha)T\log\left(1 - \frac{1}{4d\kappa}\right)} \log\left(\frac{2d\kappa\beta}{T^\alpha}\right)\right)^\alpha$$

$$= \frac{2d\kappa\beta}{T^\alpha}\left(\frac{\Delta_0}{\left(1 - \frac{1}{4d\kappa}\right)} + \left(\frac{1}{c(\alpha)\log\left(1 - \frac{1}{4d\kappa}\right)} \log\left(\frac{2d\kappa\beta}{T^\alpha}\right)\right)^\alpha\right),$$

we obtain equation 20. $\qquad\square$

In the following we also present a generalization of the above theorem to arbitrary budgets $b_t$, $\alpha > 0$ and $c$. In the context of the next theorem $B(\eta)$ (defined in the proof) is meant to be the worst case total budget required to find an $\eta$-optimizer at a given epoch, $B_T$ is the total budget and $\bar{K}(T, \eta) = \lfloor B_T / B(\eta) \rfloor$.

**Theorem D.2.** *Let $\Delta_0 \coloneqq f(x_0) - f(x^*)$, $x^*$ be the unique minimizer, $\kappa \coloneqq \beta/\mu$ be the condition number of $f$, and $\Theta \coloneqq \max_{i \in [d]} |I_k|$. Recall $c_1, c_2, c_3$ from Theorem 3.1. Let $b_T^* = \max_{t \in [T]} b_t$ and $B_T = \sum_{t=1}^{T} b_t$. Suppose that*

$$B_T \geq \max\left\{ (2d\kappa\beta)^{\frac{1}{\alpha}}, 2\frac{\alpha b_T^*}{c_3} \ln\left(\frac{\alpha b_T^*}{c_3}\right), 4\max\left\{\frac{\alpha b_T^*}{c_3}, 1\right\} \ln\left(2\max\left\{\frac{\alpha b_T^*}{c_3}, 1\right\}\right) + 2\frac{b_T^*}{c_3}\ln\left(\frac{c_2\beta\Theta\mathrm{diam}(I)}{c_1 c}\right) \right\}.$$

*If the optimizer runs ReSearch for MBCO (Algorithm 5), then, its expected optimization error is upper bounded by*

$$\left(1 - \frac{1}{4d\kappa}\right)^{\bar{K}(T,\eta)} \Delta_0 + 2d\kappa\beta\eta^2. \tag{23}$$

*Thus by setting*

$$\eta = \left( \frac{1}{c(\alpha) B_T \log\left[ \left(1 - \frac{1}{4d\kappa}\right) \left(\frac{B_T^\alpha}{2\kappa\beta d}\right)^{\frac{b_T^*}{B_T}} \right]} \log\left(\frac{2d\kappa\beta}{B_T^\alpha}\right) \right)^{\frac{\alpha}{2}}$$

*with $c(\alpha) = (\mu/(4c_1 c))^{1/\alpha}$, if $\eta > 0$[7], the expected optimization error is upper bounded by*

$$\frac{2d\kappa\beta}{B_T^\alpha} \left( \frac{\Delta_0}{\left(1 - \frac{1}{4d\kappa}\right)} + \left( \frac{1}{c(\alpha) \log\left[ \left(1 - \frac{1}{4d\kappa}\right) \left(\frac{B_T^\alpha}{2\kappa\beta d}\right)^{\frac{b_T^*}{B_T}} \right]} \log\left(\frac{2d\kappa\beta}{B_T^\alpha}\right) \right)^\alpha \right). \tag{24}$$

*Proof of Theorem D.2.* Take $\alpha, \mu, \beta, \kappa$ and $\Theta$ as defined in Theorem D.2. Let $g$ be a univariate function obtained by considering $f$ on a segment $I'$ of $I$. We can upper bound the Lipschitz constant of the function $g$ using the global Lipschitz constant $L$ of $f$, which, given that we are assuming that $x^* \in I^\circ$, can be further upper bounded by $\beta \cdot \mathrm{diam}(I)$. We start noticing that, using $b_T^* = \max_{t \in [T]} b_t$, the bound of equation 1 can be bounded from above by $2c_1 c / (\sum_{t=1}^{T} b_t)^\alpha$ whenever

$$\sum_{t=1}^{T} b_t \geq \max\left\{ 2\frac{\alpha b_T^*}{c_3}\ln\left(\frac{\alpha b_T^*}{c_3}\right), 4\max\left\{\frac{\alpha b_T^*}{c_3}, 1\right\}\ln\left(2\max\left\{\frac{\alpha b_T^*}{c_3}, 1\right\}\right) + 2\frac{b_T^*}{c_3}\ln\left(\frac{c_2 L\Theta}{c_1 c}\right) \right\}. \tag{25}$$

Indeed,

$$\frac{c_1 c}{(\sum_{t=1}^{T} b_t)^\alpha} \geq c_2 L\Theta \exp\left(-c_3 \frac{\sum_{t=1}^{T} b_t}{b_T^*}\right), \tag{26}$$

is equivalent to

$$\frac{c_1 c}{c_2 L\Theta} \geq \left(\sum_{t=1}^{T} b_t\right)^\alpha \exp\left(-c_3 \frac{\sum_{t=1}^{T} b_t}{b_T^*}\right).$$

---

[7]At a high level, note that $\eta > 0$ whenever the total budget $B_T$ is large compared to $b_T^*$.

Taking logs both sides and re-arranging this is equivalent to

$$\sum_{t=1}^{T} b_t \geq \frac{\alpha b_T^*}{c_3} \ln\left(\sum_{t=1}^{T} b_t\right) + \frac{b_T^*}{c_3} \ln\left(\frac{c_2 L \Theta}{c_1 c}\right).$$

Now consider the case when $c_2 L \Theta \geq c_1 c$, so that the second term on the right hand side is positive. So by Lemma A.2 from Appendix A of Shalev-Shwartz & Ben-David (2014), by taking

$$\sum_{t=1}^{T} b_t \geq 4 \max\left\{\frac{\alpha b_T^*}{c_3}, 1\right\} \ln\left(2 \max\left\{\frac{\alpha b_T^*}{c_3}, 1\right\}\right) + 2 \frac{b_T^*}{c_3} \ln\left(\frac{c_2 L \Theta}{c_1 c}\right),$$

equation 26 holds. On the other hand, if $c_2 L \Theta < c_1 c$, then the second term on the right hand side is negative and we can solve the stronger inequality

$$\sum_{t=1}^{T} b_t \geq \frac{\alpha b_T^*}{c_3} \ln\left(\sum_{t=1}^{T} b_t\right),$$

using Lemma A.1 from Appendix A of Shalev-Shwartz & Ben-David (2014). Thus by taking

$$\sum_{t=1}^{T} b_t \geq 2 \frac{\alpha b_T^*}{c_3} \ln\left(\frac{\alpha b_T^*}{c_3}\right),$$

equation 26 holds.

Now let $(b_t)_{t=1}^{T}$ any sequence satisfying equation 25 and let denote with $x_T$ the output of ReSearch after $T$ iterations. Notice that for any function $g$ as described above, by the strong convexity of $g$ it holds that

$$|x_T - x^*| \leq \sqrt{\frac{2}{\mu}(g(x_T) - g^*)} \leq 2 \sqrt{\frac{c_1 c}{\mu \left(\sum_{t=1}^{T} b_t\right)^{\alpha}}}.$$

Let $0 < \eta \leq \Theta$, and define $B(\eta) = \sum_{t=1}^{T(\eta)} b_t$ with $T(\eta)$ the first natural number s.t. $\sum_{t=1}^{T(\eta)} b_t \geq \left(\frac{4c_1 c}{\mu \eta^2}\right)^{\frac{1}{\alpha}}$, then it follows that the point $x_T$ found by ReSearch satisfies $|x_T - x^*| \leq \eta$. Now let $f$ be a multi-variate function satisfying the assumption of Theorem D.2. Notice that any restriction $f_k$ of $f$ along a coordinate line $k$, will also satisfies the same assumptions over the interval $I_k$ (see line 3 of Algorithm 5).

The analysis of Jamieson et al. (2012) applies to our case and they show that after $k$ epochs of coordinate descend, the expected optimization error of the current iterate is bounded above by

$$\left(1 - \frac{1}{4d\kappa}\right)^{k} \Delta_0 + 2d\kappa\beta\eta^2.$$

Recalling that $B(\eta)$ is the worst-case number budget required by ReSearch to find an $\eta$-optimizer, and if we are given a total budget of $B_T$, the number of epochs made by Line 1 of Algorithm 5 is at least $\bar{K}(T, \eta) = \lfloor B_T / B(\eta) \rfloor$. From this we get the bound of equation 23. Now notice that

$$B(\eta) \in \left[\left(\frac{4c_1 c}{\mu \eta^2}\right)^{\frac{1}{\alpha}}, \left(\frac{4c_1 c}{\mu \eta^2}\right)^{\frac{1}{\alpha}} + b_T^*\right)$$

and that

$$\eta = \left(\frac{1}{c(\alpha) B_T \log\left[\left(1 - \frac{1}{4d\kappa}\right)\left(\frac{B_T^{\alpha}}{2\kappa\beta d}\right)^{\frac{b_T^*}{B_T}}\right]} \log\left(\frac{2d\kappa\beta}{B_T^{\alpha}}\right)\right)^{\frac{\alpha}{2}},$$

with $c(\alpha) = (\mu/(4c_1 c))^{1/\alpha}$ implies the following

$$\bar{K}(T, \eta) = \lfloor B_T / B(\eta) \rfloor \geq B_T / B(\eta) - 1 = \frac{B_T}{\left(\frac{4^{r*} c_1 c}{\mu \eta^2}\right)^{\frac{1}{\alpha}} + b_T^*} - 1$$

$$= \left(\frac{1}{\log\left(1 - \frac{1}{4d\kappa}\right)} \log\left(\frac{2d\kappa\beta}{B_T^\alpha}\right)\right) - 1.$$

Replacing this into equation 23

$$\left(1 - \frac{1}{4d\kappa}\right)^{\bar{K}(T,\eta)} \Delta_0 + 2d\kappa\beta\eta^2$$

$$\leq \frac{\frac{2d\kappa\beta}{B_T^\alpha}}{\left(1 - \frac{1}{4d\kappa}\right)} \Delta_0 + 2d\kappa\beta \left(\frac{1}{c(\alpha) B_T \log\left[\left(1 - \frac{1}{4d\kappa}\right)\left(\frac{B_T^\alpha}{2\kappa\beta d}\right)^{\frac{b_T^*}{B_T}}\right]} \log\left(\frac{2d\kappa\beta}{B_T^\alpha}\right)\right)^\alpha$$

$$= \frac{2d\kappa\beta}{B_T^\alpha} \left(\frac{\Delta_0}{\left(1 - \frac{1}{4d\kappa}\right)} + \left(\frac{1}{c(\alpha) \log\left[\left(1 - \frac{1}{4d\kappa}\right)\left(\frac{B_T^\alpha}{2\kappa\beta d}\right)^{\frac{b_T^*}{B_T}}\right]} \log\left(\frac{2d\kappa\beta}{B_T^\alpha}\right)\right)^\alpha\right),$$

we obtain equation 24. Notice finally that to make the above bound valid, it has to be $B_T \geq (2d\kappa\beta)^{1/\alpha}$. $\quad\square$

# E   Adaptive Lipschitz constants

This experiment aims to show the advantages of featuring the local Lipschitz constant, as given in Theorem 3.1, over a bound that uses the global constant. To this end, we optimize the function $f(x) = -\sqrt{x} + 1$ over the interval $(a, 1)$ with $a > 0$. The Lipschitz constant of $f$ grows roughly as $1/\sqrt{a}$. Moreover, $f(x^*) = 0$ and the maximum value is smaller than 1, thus any meaningful upper bound on the optimization error should be smaller than 1. In the experiment we set $a = 0.001$, $c = 0.1$, $\alpha = 1$ (Assumption 2.1), $b_t = 1$ (for all $t$), and we let the intervals $J_t$ to be symmetric around $f(x)$ for a query at $x$ at time $t$. We run ReSearch for $T = 1000$ iterations.

Figure 6 shows that the upper bound featuring the local Lipschitz constants (local) is much tighter than that featuring the global constant (global). In particular, as denoted by the vertical lines, the former falls below 1 after 84 iterations, while the latter becomes non-trivial only at iteration 339.

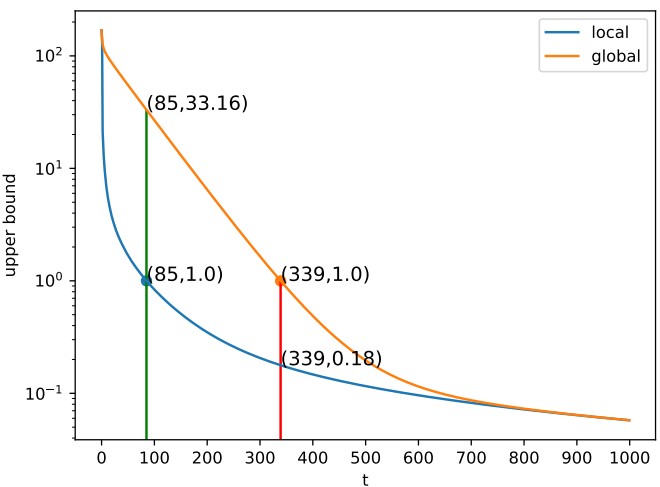

Figure 6: The vertical green line raises at the first point where the local curve falls below 1; the red line raises at the first point where the global curve falls below 1.

