# OpenReview forum: "A Theoretical Framework for Zeroth-Order Budget Convex Optimization"
_TMLR — Accepted by TMLR_

### Review · Reviewer_vNNQ · 2024-07-11

**Summary Of Contributions:**

At each time-step $t$, the optimizer can invest a budget $b_t$ in a query point $X_t$ of their choice to obtain a fuzzy evaluation of $f$ at $X_t$ whose accuracy depends on the amount of budget invested in $X_t$ across times. In the univariate case, the authors design ReSearch, an anytime parameter-free algorithm for which they argue near-optimal optimization-error guarantees.

They also present two applications of this univariate analysis. First, they used ReSearch for stochastic convex optimization. Second, they applied the d-dimensional budget problem by combining ReSearch with a coordinate descent method.

**Audience:**

No

**Claims And Evidence:**

Yes

**Requested Changes:**

- Perhaps use a different word instead of 'budget'? It is quite misleading. I think ML people use the word 'budget' as 'the amount of money available for a purpose.'

- The setting itself is not super interesting, forcing the optimizer to sample a certain spot repeatedly. I hope authors justify this setting more

    - such as papers with the same budget assumption,
    - or an example in practice (maybe univariate SCO is an interesting example, but I am not an expert in this direction and am not sure whether Agarwal et al. (2013) is really the SOTA algorithm in this field - it is more than 10 years old).

**Strengths And Weaknesses:**

Strength

1) They designed a new setting. On the traditional Zeroth order optimization, they added a budget constraint so that the learner can observe better results when he spends more budget on that observation.

2) For this problem, they designed an algorithm called ReSearch which works under minimal convexity assumptions on the target function $f$. (the environment could be adversarial)

3) They also provide a matching lower bound, even with the Lipschitz condition and prior knowledge of T and B.

4) They provide two examples to support their new setting and algorithm's practicality.

5) The algorithm is also quite straightforward - close to a binary search approach, and easy to understand.

Weakness

1) The setting that 'budget is given by environment' is not that straightforward, and some settings regarding the budget are not very convincing.

1-1) First, since there's no limitation on budget, obviously using all the budget possible is always the best strategy. Usually budget works as an overall constraint that prevents the optimizer from spending too much on one try (or observations). However, in this paper, spending less budget does not give any additional bonus, so optimizer should simply spend all their given budget for each time, which makes the problem less interesting.
1-2) Also, the width of the confidence relies on the budget spent on a 'certain point', so it naturally encourages the optimizer to choose the same point multiple times. Intuitively in practice, when someone invests a budget on a point $X_2$ close to the previous point $X_1$, then naturally we expect that the investment on $X_1$ affects the accuracy on $X_2$ too. I feel this setting is quite tailored for their algorithm ReSearch. It makes the lower bound result less surprising.

Though I know they proved a lower bound even with Lipschitz condition and known T and B, I should say the result is different from what I expected.

---

> ### Author Response · Authors · 2024-07-17
>
> 1-1) What we mean by budget $b_t$ is “the budget allocated to solve the optimization problem at time $t$”. In this sense, one indeed wants to spend it all, because it might be perishable and/or not known in advance, as in the next three examples: 1) The person who is managing the budget (in general, this is different from the person who is running the optimization) gives you some funds; these usually come with an expiration date, and you lose them if you don’t spend them all by then. 2) Consider the case of the budget being computational power coming from a machine shared by many users, such as a supercomputer; in this case, you are only allocated some time slots, and you will do your best to make the most of it as there is no upside in not using them to the fullest. 3) Your choice of lowering uncertainty in a computer experiment could be to perform 1 extra iterative step of an iterative algorithm to increase the accuracy of the computed solution. You have no direct control over how much time will be taken by the iterative algorithm, and you cannot really subdivide this time budget into smaller increments, as interrupting the run of one iteration might not produce any usable output.
>
> In addition to these considerations, note that, in our Online Protocol 1, the case where all the $b_t$’s are very small essentially models the setting where the user can choose the budget allocated to each point. Indeed, the user can select a point $X$ many times, adding small increments $b_t$, to choose the amount of budget for that point. This is discussed in the third paragraph after Theorem 3.1 and results in a smaller upper bound. This setting may not always be completely realistic, since, as we discussed above, often a perishable non-reducible budget is given to the optimizer, but it is noteworthy that our general framework covers this case, up to negligible approximations.
>
> 1-2) We agree that the one proposed by the reviewer would be an interesting generalization. Note that, if the objective is Lipschitz and the Lipschitz constant is known, then our setting allows for the learner to deduce an improved level of accuracy for neighboring points. Our results, however, are stated in a more general setting in which these assumptions might not hold. Additionally, it is unclear to us that one would improve the optimization guarantees (by more than a constant) by exploiting the information derived by the Lipschitzness of the objective. If the reviewer was thinking about something other than Lipschitzness, we are unsure if we catch their point, because, in that case, information coming from one point might not translate to any usable information for neighboring points.
>
> Requested changes
>
> - **The word “budget”**. We are open to considering alternatives if the reviewer thinks that some other words could be more clear. The way we see budgets is only slightly different from what the reviewer has in mind: instead of “the amount of money **available** for a purpose”, it is more like “the amount of money **that is allocated to be spent** for a purpose”.
>
> - **Motivating examples**. In addition to the second paragraph of the introduction, we mentioned three concrete examples in response to point 1-1. Please let us know if further clarification is needed. Regarding the one-dimensional SCO setting we consider, we confirm that the paper we mention is the SOTA (to the best of our knowledge).

---

### Review · Reviewer_LTy2 · 2024-07-28

**Summary Of Contributions:**

The paper introduces a method for zeroth-order budget convex optimization, named ReSearch. The optimizer can invest a budget at each query point to obtain a fuzzy evaluation of the function, where the length depends on the invested budget.

The authors provide near-optimal optimization error guarantees with both upper bounds and matching lower bounds.

Applications include stochastic convex optimization and a coordinate descent method for d-dimensional budget problems.

**Audience:**

Yes

**Broader Impact Concerns:**

Zeroth-order optimization is a fundamental problem in machine learning with black box models. So it would be of interest to the community.

**Claims And Evidence:**

Yes

**Requested Changes:**

It would be better if the authors could discuss more about the actual cost of the algorithms compared to existing ones in experiments.

**Strengths And Weaknesses:**

## Strengths

1. The paper is well-written and easy to follow.

2. Both theoretical and empirical results are provided. In particular, theoretical results contain both upper and lower bounds with near-optimal guarantees.



## Weaknesses

1. The proposed methods, particularly for multidimensional problems, may be complex to implement in practice.

---

> ### Author Response · Authors · 2024-08-01
>
> We start noting that a dependence on the dimension is unavoidable in zeroth-order optimization (both in deterministic or stochastic settings) (see [Nesterov2018,Shamir2013]). In particular, a linear dependence on $d$ in the optimization error is unavoidable in both settings, which implies a dependence on $d$ even in the computational complexity. Since the setting presented in the present paper is a generalization of the deterministic setting, this dependence also appears in our problem.
>
> On the other hand, our multi-dimensional method is rooted in the coordinate descent framework which is arguably among the simplest in optimization. Overall its computational complexity depends on a low degree polynomial in $d$. Indeed, Equation (20) in the appendix provides an upper bound on the optimization error as a function of time $T$. By
> equating this optimization error to a target of at most $\varepsilon$, we can show after some computations that $T$ must be of order $(d^{1+\alpha}/\varepsilon)^{1/\alpha}$. Note that $T$ is the number of  iterations of ReSearch. If the exponent $(1+\alpha)/\alpha$ can be improved, is an interesting open question we leave for future work.
>
> The following remarks are in order.
>
> - Since each iteration of ReSearch takes constant time, $(d^{1+\alpha}/\varepsilon)^{1/\alpha}$ is also the total runtime.
>
> - Since ReSearch requires constant space (it only needs to store the currently active three points) and coordinate descent requires memorizing only the current solution (a vector in $\mathbb{R}^d$) and the box boundaries ($d$ pairs of real numbers), the overall storage requirement of the method is of order $d$.
>
> - When $\alpha \rightarrow \infty$ the budget setting reduces to the deterministic setting, and the bound recovers the same linear dependence on $d$.
>
> We suggest to include a new plot (see here https://postimg.cc/Hr5JCfnX) to the paper showing the runtime of Algorithm 5 (averaged over 10 repetitions) as a function of the dimension. To parallel the experiment in the main body we set $\alpha=1$ and set the parameters to reach a target of $\varepsilon=0.1$. Notice that dealing with 100 dimensions only requires about 1 second and that the line follows the $d^2$ trend predicted by the analysis.
>
> **References**
>
> [Shamir2013] O. Shamir. "On the Complexity of Bandit and Derivative-Free Stochastic Convex Optimization". COLT 2013.
>
> [Nesterov2018] Y. Nesterov. "Lectures on convex optimization". Springer 2018.

---

### Review · Reviewer_ka7u · 2024-08-01

**Summary Of Contributions:**

This paper proposes a framework for black-box optimization of a convex function f, in which the algorithm can query the value of the function at any point x and the environment responds by announcing an interval I, which is guaranteed to contain f(x). Moreover, as the optimizer queries the same point again and again, the length of the returned interval gets smaller and smaller (see Optimization Protocol 1 and Assumption 2.1). In a way, this is like getting noisy estimates for the value of the function, where the noise decreases as you query that point more and more.

The paper proposes an algorithm ReSearch (see Algorithm 2) for the 1-dimensional case and proves a theoretical upper bound for the optimization error (see Theorem 3.1), and proves that there are cases where you cannot achieve a better bound (that is, the bound is optimal in the worst-case, see Theorem 3.2).

The authors explain how the algorithm can be deployed for stochastic convex optimization and that it improves the state-of-the-art bounds for the stochastic framework (Algorithm 4 and Theorem 4.2). They also extend the algorithm to the multidimensional case, using coordinate descent, and prove a theoretical upper bound for this case (Algorithm 5 and Theorem 4.3). They also run some experiments showing the algorithm is practical.

**Audience:**

Yes

**Broader Impact Concerns:**

Not applicable.

**Claims And Evidence:**

Yes

**Requested Changes:**

The paper is acceptable, but I have some clarification requests.

1. The way previous work is cited is weird; e.g., "The former is crucial to include scenarios where errors have long-range dependence Lahiri (2003); Beran (2017)" does not grammatically make sense.

2. Please avoid using abbreviations if possible: (a) what is DCO on page 6? (b) what is IQR in the caption of Figure 4? Please define it or give a reference.

3. In Theorem 3.2, when you write "every algorithm", does that mean "every deterministic algorithm"? Please clarify whether the bound holds for randomized algorithms as well.

4. For Theorem 4.3, you assume budgets equal 1 ... what happens if the budgets are not 1? Can you prove a guarantee in that case?

5. Footnote 3 is quite awkward and hard to understand; consider finding a way to embed it in the text (and in general, avoid footnotes as much as possible).

**Strengths And Weaknesses:**

Strengths:

1. This is a general framework that covers stochastic convex optimization as a special case, but can have other applications.
2. The proposed algorithm is natural and easy to implement, and it's nice that it has a theoretical guarantee that is worst-case optimal.
3. The paper is very well-written, full of proof sketches and ideas, useful pictures, and has enough details for a reader to understand and reproduce the results.
4. The experiments validate the robustness of the algorithm.

---

> ### Author Response · Authors · 2024-08-08
>
> 1. Thank you for spotting the typo! We should have noticed it after applying the TMLR style. We will make sure all citations are properly formatted in the revised version.
>
> 2. You are absolutely right. We will remove all these acronyms. For the record, by DCO, we meant Deterministic Convex Optimization, and by IQR, we meant interquartile range, i.e., the interval between the 25th and 75th percentiles.
>
> 3. Good question! Throughout the paper, we only considered deterministic algorithms, and we were thinking about deterministic algorithms for the lower bound. We will clarify this in the revised version. An extension of our theory to randomized algorithms is possible but more cumbersome. If the reviewer is interested, we are happy to write a full answer detailing how to do so in a separate post.
>
> 4. In Theorem 4.3, we only set $b_t=1$ for the sake of simplicity. The theorem can be extended to the case of an arbitrary budget with a simple but quite longer and more tedious analysis, in which case, the guarantee would scale with the total budget given up to time $T$.
> Please let us know if you would rather us rewrite the result and proof in general or if you believe it would suffice to add this comment. We originally chose to present it in the special case of $b_t = 1$ because we felt it would have improved readability.
>
> 5. Footnote 3 attempted to explain the "survivorship bias" that the estimators $M_t$ suffer from because of the reuse of past information. The algorithm keeps points that "look better". Hence, we expect that $\mathbb E [ M_t | X_t \text{ was kept } ] \ge f(X_t)$. We can try to rephrase the footnote and add it to the main text, or if the reviewer believes that the footnote creates more confusion than clarity, we are also OK with removing it altogether.

---

> > ### Comment · Reviewer_ka7u · 2024-08-13
> > **Re: Official Comment by Authors**
> >
> > Thanks for the responses.
> >
> > 1 and 2. Sounds good.
> >
> > 3. It's up to you whether you want to include an analysis for randomized algorithms. I don't think it's crucial.
> >
> > 4. I think adding a comment would suffice. If you want, you can include the general proof in the appendix (it's up to you).
> >
> > 5. I would recommend adding the footnote to the main body (give it a title such as "Remark."). Also, footnote 2 can also be moved to the main body, in my opinion.

---

> > > ### Author Response · Authors · 2024-08-16
> > >
> > > 3-4) We will add comments pointing out what the reviewer brought up.
> > > 5) We will move both to the main body.
> > >
> > > Thank you again for your comments and feedback!

---

### Author Response · Authors · 2024-10-03

Dear Action Editor and Reviewers,

We thank you all for your work reviewing our paper and for giving us your comments and feedback. We just updated the camera-ready version, which includes all the requested minor changes.

All the best!

The authors

---

### Decision · Action_Editor_mFq4 · 2024-09-06

**Recommendation:** Accept with minor revision

**Comment:**

The paper introduces a new setting for zero-order convex optimization where the oracle answers each query x with an interval that is guaranteed to contain f(x) and whose length decreases with the amount of budget invested on x so far. The authors propose an anytime parameter free algorithm for the univariate case and prove both an upper bound and matching (up to constants) lower bound on the optimization error. They apply their algorithm to the special case of univariate stochastic convex optimization, where they improve on existing theoretical guarantees.  They also extend the algorithm and its theoretical guarantees to the multidimensional case using coordinate descent. Some toy experiments are presented to support the theoretical claims.

Below is a summary of the strengths and weaknesses highlighted by reviewers and based on my own assessment.

Strengths:
- The proposed setting is general and includes stochastic convex optimization as special case
- The proposed algorithm is simple and easy to implement
- Theoretical guarantees are near-optimal and yield improvements on existing results in the special case of stochastic convex optimization
- Paper is well written. Results are clearly presented.

Weaknesses:
- Experiments are only done on toy synthetic problems, with small dimension d in the multivariate case.

The official recommendations from the reviewers were mixed, with two reviewers recommending to accept and one leaning to reject. The authors have addressed in their responses most reviewers' concerns. The rejection recommendation is based on the reviewer not being convinced that the setting introduced is interesting. I agree that the paper would have benefitted from more concrete examples where this setting arises. Nonetheless, I think the setting considered is reasonable, it already includes the interesting special case of stochastic convex optimization, and it is likely to have other applications. So I recommend to accept this paper with minor revision.

I request the following revisions to be made:
- Clarify if the budget is revealed before the optimizer selects a query point or after. The description given at the beginning of Section 2 seems to imply that the budget is revealed before, while the pseudocode in "Optimization Protocol 1" shows that the budget is revealed after.
- fix formatting of citations and define or remove acronyms
- clarify in Theorem 3.2 that only deterministic algorithms are considered.
- Add a comment on how Theorem 4.3 can be extended to arbitrary budget in the main text, and include the proof of this general case in the appendix.
- Include the discussion about time and space complexity of the proposed algorithm, from the response to Reviewer LTy2.
- Include the discussion clarifying the meaning of budget in the proposed setting from the response to Reviewer vNNQ.

I also recommend running experiments with larger d.

**Audience:**

yes

**Claims And Evidence:**

yes